# Rock organic carbon oxidation $CO_2$ release offsets silicate weathering sink

Jesse R. Zondervan[1,4 ✉], Robert G. Hilton[1 ✉], Mathieu Dellinger[2], Fiona J. Clubb[3], Tobias Roylands[3] & Mateja Ogrič[3]

Mountain uplift and erosion have regulated the balance of carbon between Earth's interior and atmosphere, where prior focus has been placed on the role of silicate mineral weathering in $CO_2$ drawdown and its contribution to the stability of Earth's climate in a habitable state[1–5]. However, weathering can also release $CO_2$ as rock organic carbon ($OC_{petro}$) is oxidized at the near surface[6,7]; this important geological $CO_2$ flux has remained poorly constrained[3,8]. We use the trace element rhenium in combination with a spatial extrapolation model to quantify this flux across global river catchments[3,9]. We find a $CO_2$ release of $68^{+18}_{-6}$ megatons of carbon annually from weathering of $OC_{petro}$ in near-surface rocks, rivalling or even exceeding the $CO_2$ drawdown by silicate weathering at the global scale[10]. Hotspots of $CO_2$ release are found in mountain ranges with high uplift rates exposing fine-grained sedimentary rock, such as the eastern Himalayas, the Rocky Mountains and the Andes. Our results demonstrate that $OC_{petro}$ is far from inert and causes weathering in regions to be net sources or sinks of $CO_2$. This raises questions, not yet fully studied, as to how erosion and weathering drive the long-term carbon cycle and contribute to the fine balance of carbon fluxes between the atmosphere, biosphere and lithosphere[2,11].

The tectonic activity that builds mountains results in the uplift and exposure of organic carbon (OC) that has been incorporated in rocks ($OC_{petro}$) alongside silicate mineral phases. The $OC_{petro}$ represents carbon stored in rocks that has accumulated over millions of years, previously sequestered from the atmosphere by photosynthesis and buried in sedimentary basins[12]. Indeed, sedimentary and metasedimentary lithologies presently dominate the near-surface geology of the Earth, occupying about 64% of the Earth's surface[13]; these lithologies have $OC_{petro}$ mass to mass concentration (denoted as $[OC_{petro}]$) ratios of about 0.25% to more than 1.0%, whereas igneous rocks have much lower values, effectively 0%, or in the case of some marine basalts, less than 0.1% (ref. 14).

Denudation supplies $OC_{petro}$ to the surface through physical and chemical weathering[3,15]; the rate varies with rock type, relief, tectonic uplift, climate and vegetation[16,17]. Previous work has revealed $OC_{petro}$ in soils and rivers[6,18–20] and, using data from the solid load of rivers, quantified the erosion of unweathered $OC_{petro}$[14] and its global flux at $43^{+61}_{-25}$ MtC yr$^{-1}$ (refs. 14,19). However, for weathered $OC_{petro}$, estimated global rates of $OC_{petro}$ oxidation and $CO_2$ release currently derive from carbon cycle mass balance arguments or ballpark upscaling of global river trace element fluxes[8] and have a range of estimates from 38 MtC yr$^{-1}$ (ref. 21) to 100 MtC yr$^{-1}$ (ref. 22). The uncertainty of $OC_{petro}$ oxidation fluxes is highlighted by recent work that cites a potential overall range for $CO_2$ release of 0–300 MtC yr$^{-1}$ (ref. 23).

To determine the role of rock weathering in the carbon cycle, we require a robust, global quantification of $OC_{petro}$ oxidation over Earth's surface. Here, we combine (1) a compilation of $OC_{petro}$ oxidation proxy data from dissolved rhenium (Re) in well-studied catchments around the world, (2) new probabilistic models of global $OC_{petro}$ stock and denudation and (3) a spatially explicit $OC_{petro}$ oxidation model with quantified uncertainty. This approach derives a global flux by extrapolating proxy derived $OC_{petro}$ oxidation data, while accounting for sampling bias across variables such as denudation rate and underlying geology.

## Rhenium as an $OC_{petro}$ oxidation proxy

The exploitation of the trace element Re as a proxy to study the oxidation of $OC_{petro}$ across landscapes[24,25] has been underpinned by (1) the link between OC accumulation in marine sediments and organic matter being a host of Re (refs. 26,27); (2) the paired loss of Re and $OC_{petro}$ during weathering of sedimentary rocks[7,25,28]; and (3) the geochemical behaviour of Re being exported as a dissolved oxyanion[29], flushed from a near-surface, oxidative weathering zone[25]. Studies tracking the fate of carbon released from the lithosphere during $OC_{petro}$ weathering have found it can directly enter the atmosphere as $CO_2$ (refs. 30,31) or first dissolve as inorganic carbon in water[32], and some can be incorporated into microbial biomass[8].

In this study, we compile published estimates of $OC_{petro}$ oxidation using the dissolved Re proxy, supplemented with new estimates derived from published dissolved Re concentrations[7,9,33] (Methods). A forward-mixing model is used to quantify the proportion of dissolved Re from $OC_{petro}$ oxidation using ion ratios[24,34], while constraints on the $OC_{petro}$ to Re ratio ($[OC_{petro}]/[Re]$) in weathered rocks come from new and

[1]Department of Earth Sciences, University of Oxford, Oxford, UK. [2]EDYTEM-CNRS-University Savoie Mont Blanc (USMB), Chambéry, France. [3]Department of Geography, Durham University, Durham, UK. [4]Present address: Department of Earth Sciences, University College London, London, UK. ✉e-mail: j.zondervan@ucl.ac.uk; robert.hilton@earth.ox.ac.uk

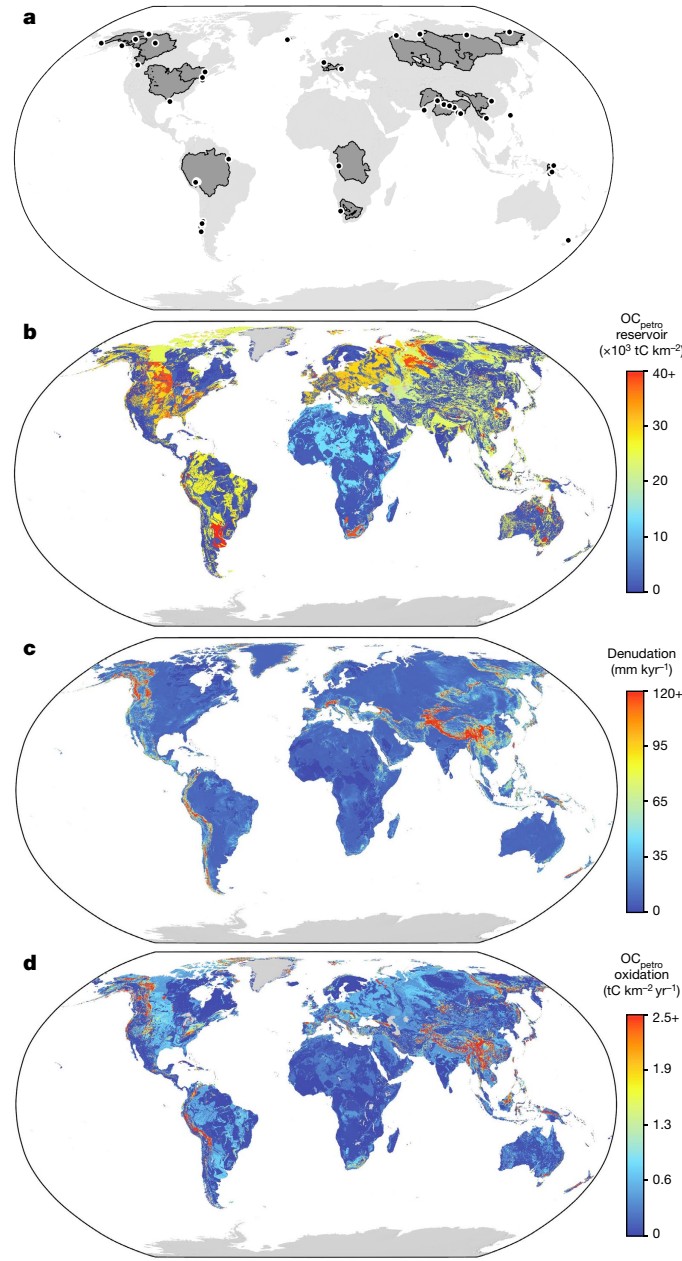

**Fig. 1 | Spatial patterns of global OC_petro stock and oxidation. a**, Locations of Re proxy samples and their upstream catchments. **b**, Spatially explicit estimates of OC_petro stocks in the upper 1 m of bedrock. **c**, Spatial model of rock denudation derived from [10]Be data and a global raster of topographic slope. **d**, OC_petro oxidation fluxes extrapolated by our calibrated spatial model over the global surface.

published measurements (Methods and Supplementary Table 3). Our compilation comprises 59 river basins, covering a range of drainage areas (50–5,900,000 km²), denudation rates and climate regimes (Fig. 1), excluding river basins with high Re pollution levels such as the Danube, Yangtze and Mississippi[9] (Methods). The total OC_petro weathering flux constrained from the Re proxy across the drainage area of river basins in the dataset is 18 MtC yr⁻¹ (17–23 MtC yr⁻¹ within one standard deviation). The river basins in this study cover 18% of the Earth's continental surface, and this flux would thus scale to $98^{+28}_{-9}$ MtC yr⁻¹ globally. However, OC_petro stocks are spatially heterogeneous, which may affect this scaling. In the next section, we obtain a robust representative total OC_petro weathering flux using a spatial extrapolation model that considers patterns in OC_petro stock and denudation rates.

## Distribution of OC_petro availability

We spatially quantify the carbon stock and weathering flux of OC_petro at Earth's surface using a data-driven modelling approach. Our model incorporates topographic and lithological factors to estimate OC_petro stocks, denudation rates and oxidative weathering rates, and is calibrated using our Re-proxy compilation (Supplementary Table 1). Unlike silicate weathering, which quickly becomes kinetically limited with increasing mineral supply by denudation[35], OC_petro weathering appears to be predominately a supply-limited process[8]. This is reflected in oxidation rates which scale with erosion up to some of the highest erosion rates found on Earth, such as Taiwan and the European Alps[7,25]. Recent work at the rock outcrop scale has shown that temperature and hydrology can control OC_petro oxidation and CO_2 release over time in locations with very high rates of denudation[30,31]. However, though the spatial control of denudation rates is well demonstrated on intercatchment OC_petro oxidation rates[7,25] our spatial catchment-scale Re-proxy compilation does not express other environmental controls (Methods). While temperature and hydrology controls likely operate, based on the available data, their spatial predictive power is small. Here, oxidative weathering is modelled at a 1-km² grid scale, resolving at the scale of catchments constrained by the Re proxy (Supplementary Table 1).

The flux of CO_2 release by OC_petro oxidation, $J_{ox}$ (mass × length⁻² × time⁻¹), can be expressed by a mass balance of the form:

$$J_{ox} = \varepsilon \times \rho \times [OC_{petro}] \times \chi \tag{1}$$

where $\varepsilon$ (length × time⁻¹) is the denudation rate, $\rho$ is rock density (mass × length⁻³), [OC_petro] is the OC concentration in rock (mass × mass⁻¹) and $\chi$ is the weathering intensity as the fraction of OC_petro weathered from rock. Weathering intensity $\chi$ has been shown to vary between low values of 0.2 in highly erosive settings[7] and very high values of 0.98 in slow denudation settings[8] with most falling in a range of 0.6–0.9 (refs. 7,33,34). Thus, $\chi$ presents a substantially smaller variance across environments in contrast to denudation rate and [OC_petro], which vary spatially by more than four orders of magnitude.

To constrain the stock of OC_petro in the near surface, we use [OC_petro] from the US Geological Survey rock geochemical database, combined with global lithological maps[13] and spatial chemical lithology classifications[36]. Our geospatial model simulates a large global near-surface OC_petro stock, with the estimate and its interquartile range at $1490^{+2580}_{-980}$ Gt OC_petro in the first metre of bedrock. This estimate is consistent with a global estimate of 1,100 Gt OC_petro within the first metre of sedimentary rocks[14], a reassessment of deep soil radiocarbon data which provides evidence for OC_petro inputs[20], and is of comparable magnitude to that of global soils (2,060 ± 215 Gt OC)[37] and marine sediments (2,322 ± 75 Gt OC)[38]. As opposed to soil OC stocks, the distribution of OC_petro is primarily controlled by the geological history of continents. While the highest [OC_petro] is found in black shales (Extended Data Fig. 4), such rocks compose a tiny fraction of the Earth's surface[13], and instead, most OC_petro is found in fine-grained sedimentary deposits such as shales (Fig. 1b). Geospatial patterns reveal low OC_petro stocks on the African continent (Fig. 1b), owing to a low occurrence of fine-grained sedimentary rocks. In contrast, substantial portions of Eurasia, South America and the middle of North America east of the Rocky Mountains contain shales. The overlap of OC_petro stocks and patterns of denudation, driven mostly by rock uplift in mountains, determines the exposure of this OC stock to oxidative weathering. We estimated denudation using a probabilistic spatial model that incorporates catchment-scale cosmogenic radionuclide (CRN) denudation rates[39], digital topography[40] and lithological maps[13]. The resultant modelled global denudation rate and its interquartile range is $11^{+13}_{-6}$ Gt yr⁻¹, within range of recent estimates of global denudation at $28^{+64}_{-20}$ Gt yr⁻¹ (ref. 16) and 15 ± 2.8 Gt yr⁻¹ (ref. 17).

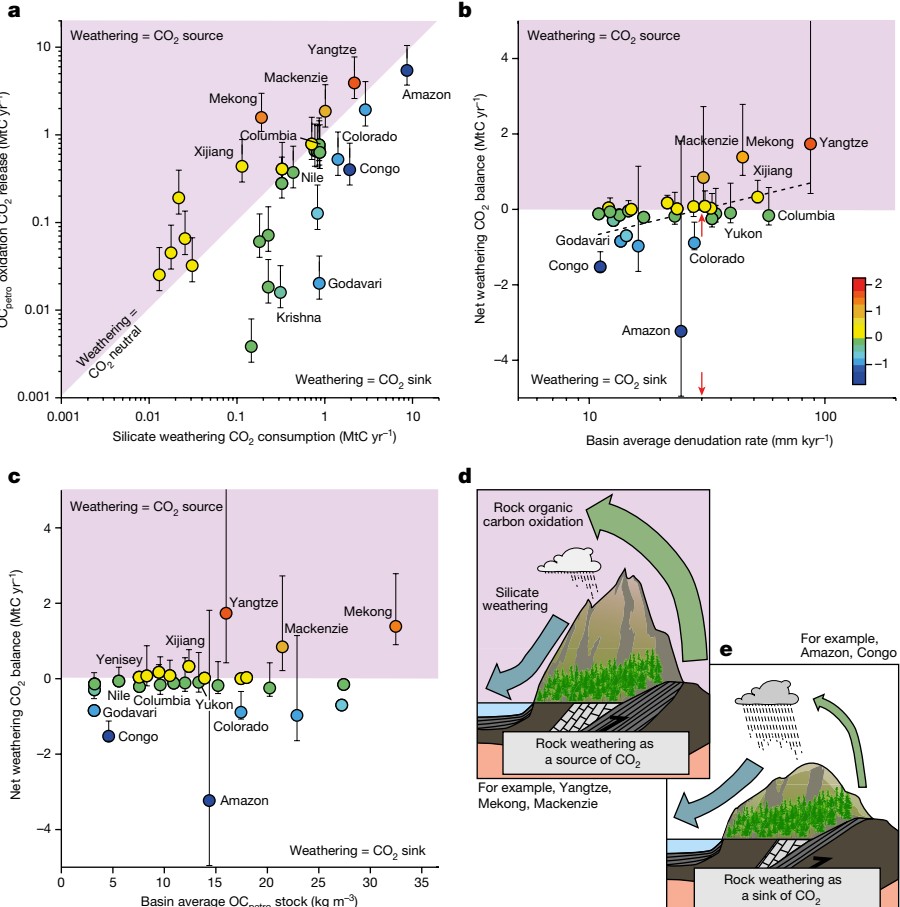

**Fig. 2 | Earth's major river basins, their silicate weathering carbon sinks and OC$_{petro}$ weathering carbon sources, and their overall rock weathering budget based on these fluxes. a**, Silicate[10] versus OC$_{petro}$ weathering fluxes and their net values. Basins that produce a net source of CO$_2$ are shown in the shaded half of the plot, with the net magnitude of the weathering CO$_2$ flux illustrated by the symbol colour (in MtC yr$^{-1}$). **b,c**, Net weathering balance versus basin-average denudation (red arrow: cross-over at about 30 mm kyr$^{-1}$)

(**b**) and versus basin-average OC$_{petro}$ stock (**c**). Error bars represent uncertainty of OC$_{petro}$ oxidation model outputs based on the uncertainty of the training data (see Methods, 'OC$_{petro}$ oxidation yields and uncertainties'). **d,e**, Variable rates of uplift and erosion, climate and OC$_{petro}$ stocks across Earth's surface impact OC$_{petro}$ and silicate weathering rates differently, leading to regions where rock weathering is a source (**d**) or a sink (**e**) of CO$_2$.

## Spatial model of OC$_{petro}$ oxidation

Rather than the more classical mean-field parametrization schemes previously employed to model OC$_{petro}$ supply rates[14], we use a probabilistic approach[41] that accounts for the uncertainties in both variables in a spatial model. In each cell, empirical probability distributions of [OC$_{petro}$] based on rock type (Extended Data Fig. 4) and probability distributions of denudation based on rock type and topographic slope (Extended Data Fig. 6) are sampled in 10,000 Monte Carlo simulations. We calibrated the geospatial model by minimizing the residuals between the modelled cell values of OC$_{petro}$ oxidation rates ($J_{ox}$) and our compilation of Re-proxy data at the river catchment scale (Methods). Thus, our approach describes the spatial patterns of oxidative weathering rate as a function of topographic slope and rock type, which leads to simulations that are consistent with an assessment of global rock nitrogen weathering patterns which are dominated by denudation of fine-grained sedimentary rocks[41].

Using our spatial model, we estimate that oxidative weathering of OC$_{petro}$ releases $68^{+18}_{-6}$ MtC yr$^{-1}$ as CO$_2$ from the land-surface environment. The flux is lower than our spatially uncorrected extrapolation of Re-proxy measurements ($98^{+28}_{-9}$ MtC yr$^{-1}$), consistent with the slight bias towards high denudation rate settings in the river basin dataset. The best estimate of the oxidative weathering flux is higher than an independent estimate of OC$_{petro}$ erosion and river transport (that is, the export of OC$_{petro}$ that has not been weathered) of $43^{+61}_{-25}$ MtC yr$^{-1}$ based on river solid load composition and flux[19], even though the uncertainties overlap. While a direct comparison of these estimates is difficult based on their quantification from dissolved versus particulate river chemistry and flux, they suggest an average weathering intensity of OC$_{petro}$ of about 60%, which is consistent with studies from large river basins[19] and intensities measured in soils[8,42].

The OC$_{petro}$ oxidation model can estimate the turnover time of OC$_{petro}$ at the surface. When combined with OC$_{petro}$ stocks, the model suggests that $0.05^{+0.12}_{-0.03}$% yr$^{-1}$ of the global OC$_{petro}$ stock in the first 10 cm of bedrock may be oxidized during denudation and weathering. A global OC$_{petro}$ loss rate of about 0.05% yr$^{-1}$ equates to a carbon turnover time (the ratio of total OC$_{petro}$ to carbon outputs by oxidation) of approximately 2,000 years. This is about double the corresponding value for global soils[43], but shorter than turnover times in tundras of approximately 3,900 years[44]. Given the large stock of OC$_{petro}$ that we report (approximately 150 Pg C in the upper 10 cm) and its turnover time, OC$_{petro}$ cannot be assumed to be inert and passive in the shallow subsurface. The input of OC$_{petro}$ into soils can also impact soil residence time estimations and lead to an underestimation of soil carbon exchange fluxes with the atmosphere[20].

Across the land surface, OC$_{petro}$ weathering is relatively focused (Fig. 1d), with variations in rock type and relief, which drive OC$_{petro}$ content and denudation, respectively, determining the magnitude

of $OC_{petro}$ oxidation and $CO_2$ release. Large regions of the African continent have lower OC stocks in bedrock and have lower relief, which together limit OC weathering. In contrast, higher $OC_{petro}$ oxidation rates are estimated for northern latitudes, where OC-rich rock and high-relief landscapes are more prevalent. Overall, 10% of the Earth surface with the highest $OC_{petro}$ oxidation rates account for 60% of the global flux in our model. The world average rate is 0.5 tC km$^{-2}$ yr$^{-1}$, hotspots (surpassing ten times world average) and hyperactive areas (all areas surpassing five times world average) are responsible for 32% and 44% of $CO_2$ emissions, respectively, while representing only 1.2% and 3% of ice-free terrestrial land area, respectively. $OC_{petro}$ weathering rates in our model are more spatially concentrated than a 1-km resolution spatial model of silicate weathering[45], where hotspots (0.51% by area) and hyperactive areas (2.9% by area) accounted for 8.6% and 19.6% of total $CO_2$ consumption, respectively. This outcome appears reasonable because $OC_{petro}$ is less common spatially than silicate minerals and reacts faster[3,25].

## Weathering CO₂ sources versus sinks

The $OC_{petro}$ weathering flux and release of $CO_2$ to the atmosphere of $68^{+18}_{-6}$ MtC yr$^{-1}$ is similar to global terrestrial $CO_2$ uptake by silicate weathering (94–143 MtC yr$^{-1}$)[10]. Silicate weathering involves dissolved and gaseous $CO_2$ uptake through bicarbonate production and the release of dissolved ions, some of which then precipitate as marine carbonate rocks[4]. The resultant total transfer of carbon from the atmosphere to the lithosphere by silicate weathering is 47–72 MtC yr$^{-1}$. Besides their opposing impacts on the transfer of carbon between the atmosphere and lithosphere, fluxes of silicate weathering versus $OC_{petro}$ oxidation may have contrasting responses to climate. Silicate weathering is invoked as negative feedback to climate warming through increased rates of silicate weathering from increased temperature and a more vigorous hydrological cycle, drawing down more $CO_2$ (refs. 35,46). In contrast, in high denudation rate settings the $CO_2$ release from $OC_{petro}$ oxidation may increase with temperature[30,31], while links to glacial erosion processes complicate the feedback between oxidative weathering and climate[33].

Silicate and $OC_{petro}$ weathering processes may overlap, as sedimentary rocks contain silicate minerals as well as $OC_{petro}$; however, the relative magnitude of these fluxes will vary spatially with climate, rock type and denudation[35,46]. We assess the net balance of rock weathering within major river basins (Fig. 2), using our $OC_{petro}$ oxidation model and silicate weathering estimates[10].

Within uncertainties, rock weathering in about a third of the major river basins is a net source of $CO_2$ after $OC_{petro}$ oxidation is considered, even while using the values of initial atmospheric $CO_2$ consumption of silicate weathering rather than the smaller quantity of $CO_2$ eventually locked up in the lithosphere through carbonate precipitation of the associated released dissolved ions (Fig. 2a). The Yangtze (Changjiang) and Mekong draining the eastern flanks of the Himalayas and the Mackenzie River draining shales west of the Rockies in Canada are major sources of $CO_2$ from rock weathering. These high-emitting basins have in common some of the highest basin-average denudation rates and $OC_{petro}$ stocks (Fig. 2b,c), which is consistent with $OC_{petro}$ oxidation being driven by $OC_{petro}$ stocks and denudation (equation (1))[7,8,25].

Hotspots of $CO_2$ release during rock weathering appear to lie at the edges of major active mountain ranges where relatively young, marine sedimentary deposits are uplifted and supplied to the oxidation process through denudation. Examples include the shales of the Himalayan collision zones and east of the Rocky Mountains (Fig. 1b,d). On the other hand, basins where rock weathering is the biggest net sink of $CO_2$ do not necessarily lie at the extremes of low denudation and low $OC_{petro}$ stocks. Though the tropical Congo River and volcanics-dominated Godavari River basins have low basin-average denudation rates and low $OC_{petro}$ stocks, neither one is the biggest weathering sink; that

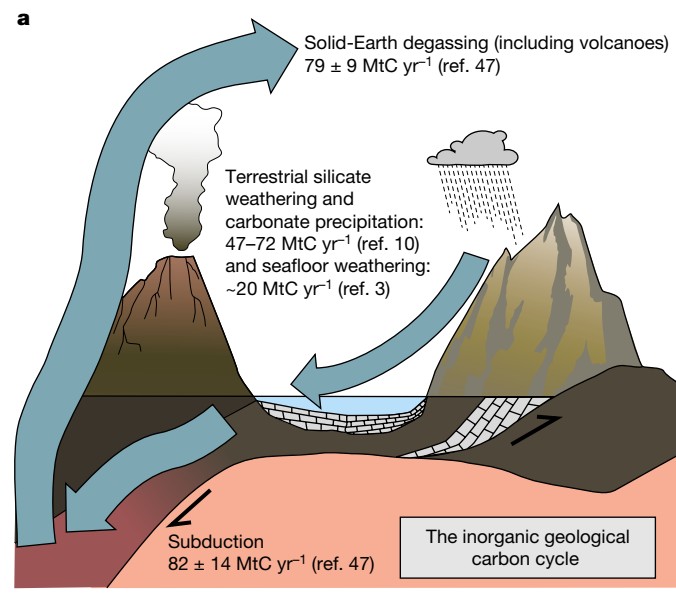

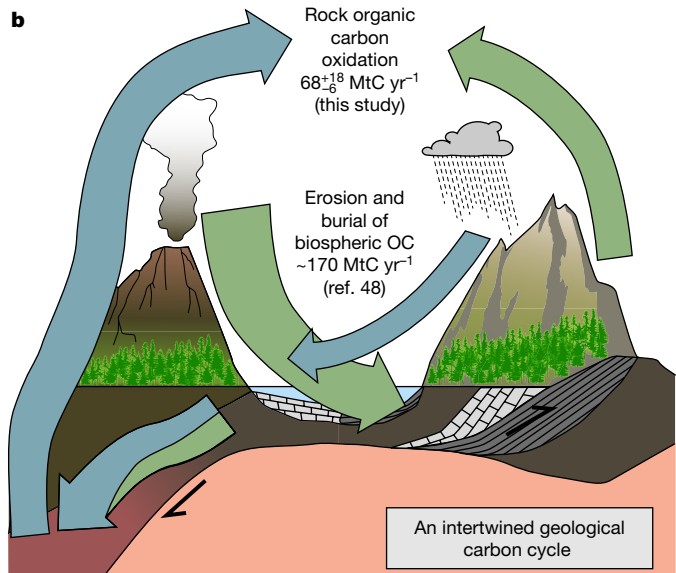

**Fig. 3 | A shift in understanding the geological carbon cycle. a**, The inorganic geological carbon cycle relies on a global balance between solid-Earth $CO_2$ degassing and silicate weathering. **b**, The emerging understanding of the role of organic carbon in the global geological carbon cycle, supported by the high flux of $OC_{petro}$ oxidation reported in this study. Hence, the biological, chemical and physical processes of biospheric OC production, burial and release control long-term climate variability and stability.

distinction applies to the Amazon River basin, which lies in the global middle range of denudation rates and $OC_{petro}$ stocks (Fig. 2b,c). There, the kinetically limited silicate weathering reaction benefits from long sediment residence times and a warm, humid climate.

While the Andes is a hotspot for $OC_{petro}$ oxidation fluxes (Fig. 1d), the exceptionally large lowland drainage area of the Amazon means that $OC_{petro}$ oxidation may be supply limited. In a third of river basins weathering remains carbon neutral within uncertainty; for example, such is the case with the volcanic-rich Columbia River catchment.

To avoid large swings in atmospheric $CO_2$ over millions of years and maintain an apparent close balance of $CO_2$ sources and sinks[2,11], any potential imbalances in weathering-derived carbon fluxes must be addressed by accounting for other components in the long-term carbon cycle. Solid-Earth degassing associated with volcanoes and diffuse release from metamorphism in subduction zones is responsible for

$79 \pm 9$ MtC yr$^{-1}$ released into the atmosphere (Fig. 3a)[47], while any additional (non-subduction) global $CO_2$ release during orogenic metamorphism and sulfide oxidation and inorganic C uptake during seafloor weathering are more poorly constrained[3]. As our results show that the weathering of $OC_{petro}$ offsets silicate weathering in the long-term carbon cycle, a large additional sink of $CO_2$ is needed. This may be provided by burial of organic matter in ocean sediments, which could contribute as much as 170 MtC yr$^{-1}$ (Fig. 3b)[48]. In addition, as $OC_{petro}$ fluxes can overtake silicate weathering during periods of more intense uplift and erosion (Fig. 2b,d and Extended Data Fig. 8), the question whether orogenic periods in Earth history are sources or sinks of atmospheric $CO_2$ is now a reopened question[3,31,49,50]. The net balance will depend on factors such as transport of terrestrial biospheric carbon to oceans ($157^{+74}_{-50}$ MtC yr$^{-1}$)[19] and its burial[19]. A global comparison of catchment-scale $OC_{petro}$ oxidation yields and estimated terrestrial biospheric OC burial (Extended Data Fig. 8) suggest the OC burial can apparently offset or even overcompensate $CO_2$ release from $OC_{petro}$ oxidation. This understanding persists when the additional marine OC burial sink in sediment is factored into global flux estimates (Fig. 3b)[48]. The dynamics of Earth's weathering thermostat thus need to be revisited to account for variation in all these fluxes and consider how their relative importance may have changed as life evolved and the OC stocks of sedimentary rocks have increased[3,22].

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

## Methods

The workflow of materials and methods (Extended Data Fig. 1) starts with the compilation and derivation of rhenium-concentration-based $OC_{petro}$ oxidation flux estimates, discussed just below. Then we detail the 'Global spatial $OC_{petro}$ oxidation model' and its application of sub-models for $OC_{petro}$ stocks and denudation, and the Monte Carlo routines used in the model's approach. Finally, we discuss the 'Limitations and uncertainties' involved in our methods and calculations.

### Rhenium-based river catchment estimates of $OC_{petro}$ oxidation

From a series of dissolved rhenium measurements (typically completed by ICP-MS), the dissolved Re flux $J_{Re}$ (t yr$^{-1}$) can be used to estimate $OC_{petro}$ oxidation flux, $J_{OCpetro-ox}$ (tC yr$^{-1}$) using:

$$J_{OCpetro-ox} = J_{Re} \times \left( \frac{[OC_{petro}]}{[Re]} \right)_i \times f_c \qquad (2)$$

where $f_c$ is the fraction of dissolved Re derived from $OC_{petro}$ oxidation[34] and $([OC_{petro}]/[Re])_i$ is the organic carbon to rhenium concentration ratio (g g$^{-1}$) in rocks of type $i$ undergoing weathering. In some catchments where it may be important, an additional term, not shown in equation (2), has been applied to correct for the presence of graphite, which may not undergo alteration during weathering[33].

**Compiled published measurements.** In this study we compile estimates of $OC_{petro}$ oxidation using the dissolved Re proxy from published literature (Supplementary Table 1). These include the Yamuna River, India[24]; ten Taiwanese rivers[7]; four rivers from the western Southern Alps[33]; four rivers from the Mackenzie Basin, Canada[3,34]; and six rivers draining the Peruvian Andes[51]. Two Swiss catchments[25] are not included because of their very small catchment area compared to the geospatial scales over which we complete the upscaling.

For some of these case studies, dissolved rhenium flux has been estimated from repeated sampling and discharge records[34], while earlier studies all include single snapshot samples[7,24], and all include measurements of the local sedimentary rock composition. Most of these compiled studies have used dissolved ion ratios to estimate the source of dissolved Re, akin to $f_c$ (equation (2)), apart from the Taiwan dataset[7]. While uncertainties on the $OC_{petro}$ oxidation yields appear relatively large (Supplementary Table 1), it is important to note that the measured range in yields is much larger than the uncertainties.

**New estimates of $OC_{petro}$ oxidation.** To expand the 25 estimates of $OC_{petro}$ oxidation from river catchments described previously, we build on a previous study of dissolved Re fluxes in large rivers that reports dissolved Re concentrations and fluxes for major basins around the world[9]. We use these measurements and combine them with estimates of $f_c$ and $([OC_{petro}]/[Re])_i$, discussed in the following sections, to calculate $OC_{petro}$ oxidation yields with associated uncertainties using published approaches[25,34].

In locations with substantial local sources of fossil fuel combustion (for example, coal-fired power plants or steel works), rainwater can contain concentrations of Re that approach those of river water[8,52], whereas locations that have minimal impacts from local pollution sources have Re concentrations in rainwater that are below detection[25,33]. In the large river dataset[9], some large rivers are noted to have markedly increased Re concentrations and fluxes; the conclusion is that this was due to anthropogenic Re inputs. In a first-order catchment of the Mississippi Basin, this has been confirmed by a detailed Re mass balance[52]. A study of Re across Indian catchments suggests that while Re in Himalayan catchments and the mainstem Ganges and Brahmaputra behave conservatively, peninsular lower relief catchments with denser populations and industrial activity suggest anthropogenic inputs[53]. For this study's purpose, to quantify weathering reactions, we only use Himalayan rivers and the mainstem Ganges and Brahmaputra in India, and we have further excluded Re data from the Danube, Mississippi and Yangtze rivers from our analysis. Our addition of catchment Re data to the Miller dataset includes a large contribution of small upland catchments with higher average erosion rates, where the authors of these studies selected sites with minimal human disturbance (Supplementary Table 1). We further consider the role of anthropogenic Re in our model results in the 'Limitations and uncertainties' section.

**Estimation of Re source and $f_c$.** To estimate the fraction $f_c$ of dissolved Re sourced from $OC_{petro}$ for the rivers in the Re flux dataset[9], we follow a previously used forward model mixing approach[25,34]:

$$Re_{OC} = Re_{tot} \times f_c = Re_{tot} - Re_{sulf} - Re_{sil} \qquad (3)$$

where $Re_{OC}$ is the rhenium concentration of $OC_{petro}$-derived Re in the dissolved load, $Re_{tot}$ is the measured Re concentration, $Re_{sulf}$ and $Re_{sil}$ are the concentrations derived from weathering of sulfide and silicate minerals, respectively. These unknowns can be quantified as:

$$Re_{sulf} = [SO_4] \times \left( \frac{[Re]}{[SO_4]} \right)_{sulf} \qquad (4)$$

$$Re_{sil} = [Na] \times \left( \frac{[Re]}{[Na]} \right)_{sil} \qquad (5)$$

where the element ratios of the end members for silicate, $([Re]/[Na])_{sil}$, and sulfide, $([Re]/[SO_4])_{sulf}$, are defined, and with the assumption that the dissolved sulfate ($SO_4$) and sodium (Na) respectively derive only from sulfide oxidation and silicate weathering and are conservative. This returns an upper bound on the $Re_{sulf}$ and $Re_{sil}$ components (Supplementary Table 2). Following recent work[34], we use a range of values for each, where $([Re]/[SO_4])_{sulf}$ ranges from $0.2 \times 10^{-3}$ to $4 \times 10^{-3}$ pmol µmol$^{-1}$ (ref. 23) and $([Re]/[Na])_{sil}$ ranges from $0.4 \times 10^{-3}$ to $2 \times 10^{-3}$ pmol µmol$^{-1}$. Here, we correct Na$^+$ concentrations for atmospheric-derived Na, $[Na^+]*$, where $[Na^+]* = [Na^+] - [Cl^-] \times 0.8$, assuming all Cl$^-$ derives from precipitation, which has a molar $[Na^+]/[Cl^-]$ ratio of 0.8. We similarly correct for atmospheric $SO_4$ inputs.

**Constraints on $([OC_{petro}]/[Re])_i$.** A recent compilation[54] provides measurements of $([OC_{petro}]/[Re])_i$ from rock samples of different ages around the world. However, most of these measurements were made on black shales with $OC_{petro}$ contents greater than 1%, which occur on only 0.3% of the Earth surface (Extended Data Fig. 5). Riverbed material sediments from erosive catchments provide an alternative way to capture landscape-scale average rock composition, albeit with some potential for weathering to alter the primary signal. Here we compile measurements of $[OC_{petro}]/[Re]$ on bed materials from rivers around the world (Supplementary Table 3 and Extended Data Fig. 2) and supplement this dataset with additional samples from mudrocks of the Eastern Cape, New Zealand, and the Peruvian Andes measured using methods described previously[25]. We find that regions with lower $OC_{petro}$ concentrations that are more typical of sedimentary rocks at the continental surface—units including fine-grained sedimentary rocks that make up more than 35% of Earth surface (Extended Data Fig. 5)—have lower and more consistent ratios of OC and Re in their rocks. The samples from the Peel River in the Mackenzie River basin[34] overlap the lower end of the published black shale values. Since this is the catchment with the highest proportion of black shales in our Re dataset, these samples allow us to capture the imprint of this important marginal lithology at the landscape scale.

The bedrock composition in the catchments of rivers studied in the Re flux dataset[9] is not reported. However, we note the good geographic coverage and number of samples that we have from riverbed materials

from erosive settings around the world. These provide constraint on the initial OC to rhenium ratio in the rocks. To conservatively quantify uncertainty in the range of $OC_{petro}$ oxidation rates from dissolved-Re data, we perform a Monte Carlo simulation in which we uniformly sample the entire range of measured $[OC_{petro}]/[Re]$ values, from low values indicative of carbon-poor and/or metamorphic rocks $2.5 \times 10^{-8}$ g g$^{-1}$ (ref. 33) towards $1.26 \times 10^{-6}$ g g$^{-1}$ (ref. 34) in catchments with higher OC in rocks (Supplementary Table 3 and Extended Data Fig. 3).

**$OC_{petro}$ oxidation yields and uncertainties.** Equation (2) is used for each basin in the Re flux dataset[9]. Uncertainties in $f_c$ derive from the range of values used in the sulfide and silicate end member compositions (equations (4) and (5)). For $[OC_{petro}]/[Re]$, we use the range of values discussed in the just-previous section on constraints (Extended Data Fig. 3). A Monte Carlo uncertainty propagation is used on these variables, with 10,000 randomly selected combinations of input values (with uniform sampling) are used to estimate $J_{OCpetro-ox}$ for each basin. The median value of the Monte Carlo simulation and the interquartile range are reported (Supplementary Table 1).

**Geospatial catchment boundaries.** To derive the catchment outlines and areas corresponding to the Re-proxy samples in our compiled dataset, we used the HYDROSHEDS flow direction grid at 3 arc-second resolution[55] and ArcGIS Pro[56]. Catchments outside the latitudinal cover of HYDROSHEDS were derived from the HYDRO1K flow direction grid[57] and catchments in Iceland were derived from ALOS AW3D using TauDEM functionality in OpenTopography[58]. While most published sample coordinates (Supplementary Table 1) give the correct location on the cited drainage systems, in a handful of cases, coordinates had to be amended by up to a few kilometres, which may reflect errors in transcribing (for example, Kikori and Purari[9]). Final quality control included a comparison of the extracted drainage basin areas and those published, with good agreement overall (less than 2% residual). However, some drainage areas cited in the Re flux dataset[9] refer to the river mouth, rather than the river catchment upstream of the Re sample location. In these cases, we use the Re sample location and its upstream catchment. Finalized coordinates of Re samples determined for each drainage system, with the corresponding upstream drainage area, are given in Supplementary Table 1. Spatial files of upstream drainage boundaries and Re sample locations are available on Zenodo (available from https://doi.org/10.5281/zenodo.8144244). To convert dissolved Re concentrations into Re fluxes, average annual water discharge was calculated using published numbers at gauges (Supplementary Table 1) and scaled to the upstream drainage area of Re sample locations.

In addition to spatial catchment boundaries for the Re proxy dataset, we compare our spatial model output to published estimates of silicate weathering[10] that use the GRDC dataset in *Major River Basins of the World*[59]. Drainage areas used by ref. 10 have slight discrepancies with those found in the GRDC dataset. We account for these in our analysis of major river basin net weathering flux (Supplementary Table 4).

## Global spatial $OC_{petro}$ oxidation model
In the following three sections, we provide additional rationale and details of the modelling approaches. The model procedures apply two spatial probabilistic subroutines; one deals with $OC_{petro}$ stocks in surface rocks and the other with spatially defined denudation rates. These are combined in a Monte Carlo framework alongside the Re-proxy river catchment data to optimize the model and then extrapolate $OC_{petro}$ oxidation rates (Extended Data Fig. 1). Model simulations were implemented at 1-km grid scale (Mollweide projection, WGS84 datum) in the Python programming language[60].

**$OC_{petro}$ stocks.** Rock samples from the USGS Rock Geochemical Database, sorted into lithological categories (Supplementary Table 5), were mapped onto units of the highest-resolution global lithological maps currently available[13]. Extended Data Fig. 4 shows the $OC_{petro}$ concentration of key lithologies in the USGS Rock Geochemical Database. Weight percentage values from the USGS Rock Geochemical Database were converted to $OC_{petro}$ stock using rock densities (Supplementary Table 5). In our Monte Carlo framework, $OC_{petro}$ stocks at each grid cell were sampled independently using the empirical distributions of rock $OC_{petro}$ content derived from both the USGS database (Extended Data Fig. 4) and our unit classification (Supplementary Table 5). In our lithology model, complex mapped units present in GLiM consist of a combination of carbonates and silicates of various grain sizes (Extended Data Fig. 5 and Supplementary Table 5). To calculate the $OC_{petro}$ reservoir among these units, we derive the fractional abundance of lithology types ($F_n$) from continental-scale literature estimates[36]:

$$[OC_{petro}]_{rock} = F_1([OC_{petro}]_{lithology,1}) + F_2([OC_{petro}]_{lithology,2}) + \cdots$$
$$+ F_n([OC_{petro}]_{lithology,n}) \tag{6}$$

$$F_1 + F_2 + \cdots + F_n = 1 \tag{7}$$

**Denudation model.** The denudation model is parametrized using a regression approach, similar to techniques applied elsewhere[16,41]. We regressed a compilation of long-term catchment-scale $^{10}$Be denudation estimates[39] against mean local slope generated from the Geomorpho90m dataset[40]. Mean local slope was calculated using the focal statistics tool in ArcGIS Pro[56] and the Geomorpho90m slope dataset with a 5-km moving radius. Slope values were then matched to $^{10}$Be denudation estimates at a single cell based on the reported longitude and latitude. A quantile regression approach[41,61,62], allows us to mitigate over- and underestimations inherent in using a mean model fit to the global land surface[16] (Extended Data Fig. 6). For each unique slope value in the global raster, denudation quantiles were used to construct a cumulative distribution function which could be sampled in each Monte Carlo run (compare ref. 41).

We account for differential erodibilities of sedimentary, crystalline metamorphic and igneous rock types by running regressions between slope values and $^{10}$Be values for each rock type (Extended Data Fig. 6). Thus, only $^{10}$Be values from catchments dominated (more than 80%) by one rock type are used in this regression. This accounting of erodibilities is important, as OC-rich shales are weaker and more erodible than OC-poor strong igneous rocks. Residuals between the CRN denudation dataset and the modelled denudation do not change when differential rock erodibility is considered. However, when combined with our $OC_{petro}$ stock model, the rock erodibility-corrected OC supply rate model results in 20% higher rates. We also consider the grid-scale bias considered by previous workers[16,41]: as DEM resolution decreases, slope—as the spatial derivative of elevation—decreases, resulting in an artificial flattening effect[16]. As our Monte Carlo framework is computationally intensive, using a 90-m-resolution global raster input would not be feasible. However, we use a 90-m-resolution slope dataset to run regression curves as shown in Extended Data Fig. 6, after which we output a 90-m-resolution raster dataset of estimated denudation rates using the median regression curve. By resampling the raster dataset of estimated denudation rates to 1-km resolution after conversion from slope values, we avoid the bias that can lead to an underestimation of denudation by the flattening effect. In our Monte Carlo framework, the quantile regression curves for each raster value can then be sampled to draw a representative denudation value out of the empirical distribution of denudation rates.

**Model calibration.** The global model is calibrated by minimizing the residual with the Re-proxy-based estimates of $OC_{petro}$ oxidation (tC yr$^{-1}$) from 59 globally distributed river basins (Supplementary Table 1). Model selection was performed by running a Monte Carlo simulation

(10,000 runs), using the variable $OC_{petro}$ stock and denudation models described above, to find the output which minimizes total residuals across all 59 calibration basins simultaneously, such that the sum of all basin residuals was less than 1%. These simulations were run on the University of Oxford's Advanced Research Computing (ARC) facility, taking about 24 core hours per simulation. The residuals of individual basins can be quite large for the biggest catchments (for example, the Amazon basin), but the relative residual, especially for the larger basins, falls within the uncertainty of model outputs, while accurately predicting the total $OC_{petro}$ oxidation flux globally (Extended Data Fig. 7). We note that, overall, in basins with moderate $OC_{petro}$ oxidation fluxes, the model may return conservative estimates. However, because this model has the advantage of being globally and spatially explicit, regional over- and underestimation of $OC_{petro}$ oxidation found mostly at a local scale (less than $10,000 km^2$) tend to trade off while we are able to capture larger regional differences due to tectonics and geological setting (Fig. 1d).

## Limitations and uncertainties

There is a temporal mismatch between the CRN denudation data that inform our probabilistic denudation model, and our Re-proxy calibration data. The Re-proxy-based $OC_{petro}$ oxidation fluxes used to calibrate our spatial extrapolation model capture fluxes from global rivers within the past decade or less. The CRN technique integrates denudation fluxes that span a millennium or more. Anthropogenic land-use change has doubled erosion and weathering since the early 1900s (ref. 63); hence, our global scale estimates of $OC_{petro}$ oxidation rates reflect the combined influence of natural and anthropogenic activities on global weathering rates, which cannot be deconvolved in this present study.

Results of model versus Re-predicted $OC_{petro}$ oxidation fluxes help us assess the potential for anthropogenic Re input to impact our estimates (Extended Data Fig. 7). We have considered anthropogenic Re inputs by removing three large river basins from a previous compilation[9] and by adding carefully selected river catchment sites to our Re dataset (see the Methods section headed 'Rhenium-based river catchment estimates of $OC_{petro}$ oxidation'). In addition, our conversion of Re fluxes to $OC_{petro}$ oxidation is conservative because we uniformly sample the range of Re/OC ratios starting at the lowest measured Re/OC ratio (see the Methods section headed 'Constraints on ([$OC_{petro}$]/[Re])'). This leads to error bars within our estimates that are conservatively large. Most notably, the model outputs of $OC_{petro}$ oxidation versus the Re-estimated fluxes for each basin (Extended Data Fig. 7) show a tendency for the model to underpredict smaller catchments more than larger catchments. Our confidence in the weathering signal from Re in the small upland catchments is highest, and the upland, high-erosion-rate regions that these catchments sample contribute a dominant proportion of the global $OC_{petro}$ flux in our model. While we cannot completely deconvolve the effect of anthropogenic Re in our constraints, we have confidence that the effect is unlikely to result in a significant overprediction of global $OC_{petro}$ flux estimates.

The global extrapolation of $OC_{petro}$ oxidation proxy data attempts to account for the dataset's underlying heterogeneities in denudation and $OC_{petro}$ stocks. However, it does not consider variability in temperature or precipitation which may control weathering—as seen in small-scale field measurements of $OC_{petro}$ oxidation at sites of high denudation[30,31]. This is primarily due to the size of the Re-proxy catchment database, its spatial coverage and uncertainties inherent in any proxy approach. While the Re-proxy dataset is latitudinally variable (Fig. 1a) the model misfit minimization procedure shows the first-order controls on flux by $OC_{petro}$ stock and denudation (Extended Data Fig. 7), meaning that any climatic controls on weathering could be not resolved at the global scale. We note that any bias introduced by extrapolating the global Re-proxy data without including climatic spatial controls on weathering intensity is likely to be minimal, because the underlying dataset spans from the tropics to Arctic locations.

Previous work has suggested that $OC_{petro}$ oxidation and cycling of other OC pools can take place during floodplain transport in large fluvial systems[64,65]. While the Re-proxy dataset includes large basins with extensive floodplain areas (Fig. 1a), our model–data misfit approach may attribute the downstream fluxes to any higher denudation parts of the catchment. When the model is then upscaled, in lowland floodplain areas where fluvial processing and recycling of sediment[65] can allow biogeochemical reactions to continue[65], we may predict conservative $OC_{petro}$ oxidation rates. At present, we lack empirical data to partition weathering between mountain and floodplain sections[3]. An alternative way to view this is that the removal of alluvial domains from contributing to denudation (since these are depositional) holds minimal control on the overall estimate (less than 1%). This result comes from a comparison of model outputs under two parametrization schemes: one where denudation occurs over all ice-free lands versus one where denudation only occurs in ice-free non-alluvial landscapes. Overall, the model's largest contributor to uncertainty is in the conversion of dissolved Re fluxes to $OC_{petro}$ oxidation estimates, which are extrapolated in our spatial model. This conversion depends on [$OC_{petro}$]/[Re] ratios which introduce most of the uncertainty in the resulting $OC_{petro}$ oxidation rates (see the Methods section '$OC_{petro}$ oxidation yields and uncertainties') and therefore in the model's global output. More constraints on the relative 'grey shale' versus 'black shale' contribution to catchment Re fluxes could help tighten uncertainty in [$OC_{petro}$]/[Re] ratios (see ref. 66 for a discussion).

Finally, our model includes implicit assumptions and features of the datasets which must be acknowledged. First, the model assumes a steady state, which might not accurately describe $OC_{petro}$ oxidation in regions responding to changes in uplift, deglaciation or human activities, which may not yet have reached steady-state conditions.

Second, most catchment-scale CRN denudation data used in our model derive from lithologies that are quartz-rich and coarse-grained. These typically have lower erodibilities, potentially leading to an underestimation of denudation rates of softer shales which contain the majority of $OC_{petro}$ stocks.

## Data availability

All dissolved rhenium sample data are available in Supplementary Tables 1–5, in addition to which geospatial data, including those for Fig. 1, are deposited in a Zenodo repository (https://doi.org/10.5281/zenodo.8144244). Source data are provided with this paper.

## Code availability

The spatial $OC_{petro}$ oxidation model contains two spatial probabilistic subroutines of (1) $OC_{petro}$ stocks in surface rocks and (2) spatially defined denudation rates. These are combined in a Monte Carlo framework alongside the Re-proxy river catchment data to optimize the model and then extrapolate $OC_{petro}$ oxidation rates (Extended Data Fig. 1). Model simulations were implemented at 1-km grid scale (Mollweide projection, WGS84 datum) in the Python programming language. R and Python code, their environments and the necessary data files to run these are all deposited in a Zenodo depository (https://doi.org/10.5281/zenodo.8144244) and additionally code is available from GitHub: https://github.com/jessezondervan/Global_OCpetro_Oxidation.

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

**Acknowledgements** We would like to acknowledge the use of the University of Oxford Advanced Research Computing (ARC) facility in carrying out this work (https://doi.org/10.5281/zenodo.22558). In addition, this work is based on digital elevation products and high-performance topographic analysis services provided by the OpenTopography Facility with support from the National Science Foundation under NSF Award Numbers 1948997, 1948994 and 1948857. This study was supported by the European Research Council Starting Grant ROC-CO2 678779 (R.G.H.).

**Author contributions** J.R.Z. and R.G.H. conceptualized the study. J.R.Z., R.G.H. and F.J.C. developed the methodologies. J.R.Z. and R.G.H. conducted the investigation. R.G.H., M.D., T.R. and M.O. acquired the resources. Data visualization was done by J.R.Z. and R.G.H., while J.R.Z., R.G.H. and M.D curated the data. R.G.H. acquired the funding. J.R.Z. and R.G.H wrote the original draft of the paper. The paper was reviewed and edited by all the authors.

**Competing interests** The authors declare no competing interests.

**Additional information**
**Correspondence and requests for materials** should be addressed to Jesse R. Zondervan or Robert G. Hilton.

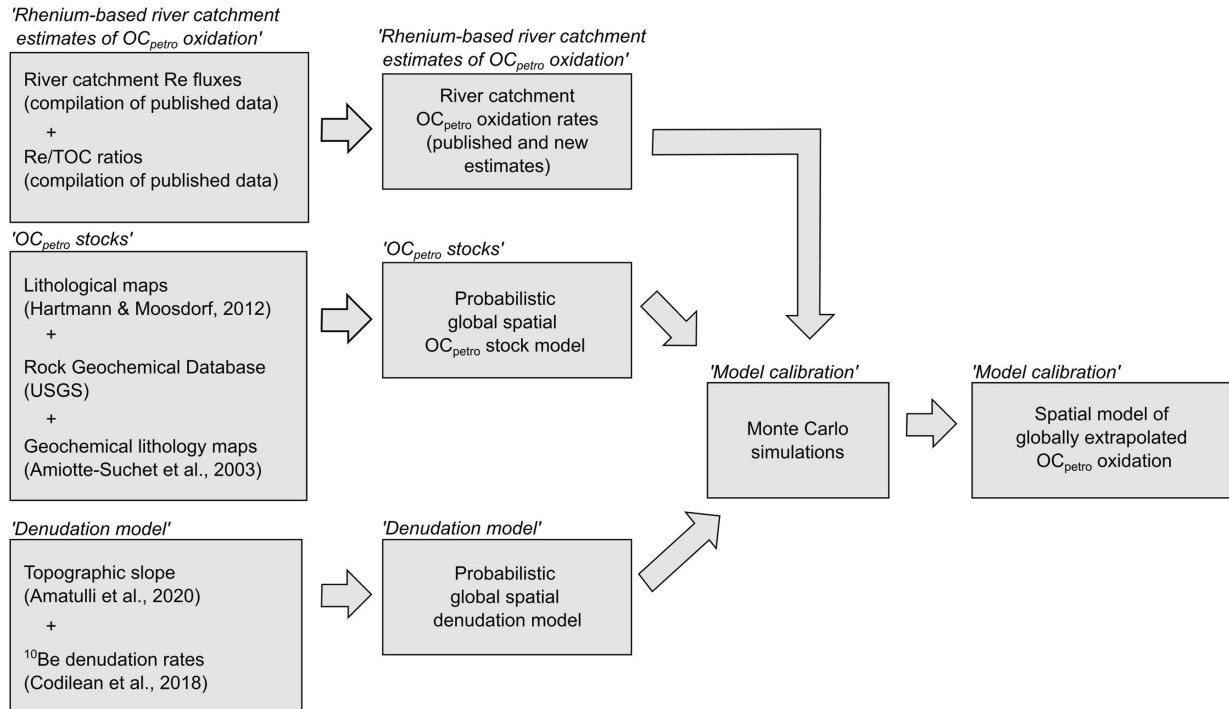

**Extended Data Fig. 1 | Flow chart of data, subroutines and outcomes and the relevant Methods sections.** Probabilistic geospatial models of OC_petro stocks and denudation are used to extrapolate a global compilation of river catchment OC_petro oxidation rates across the Earth's surface.

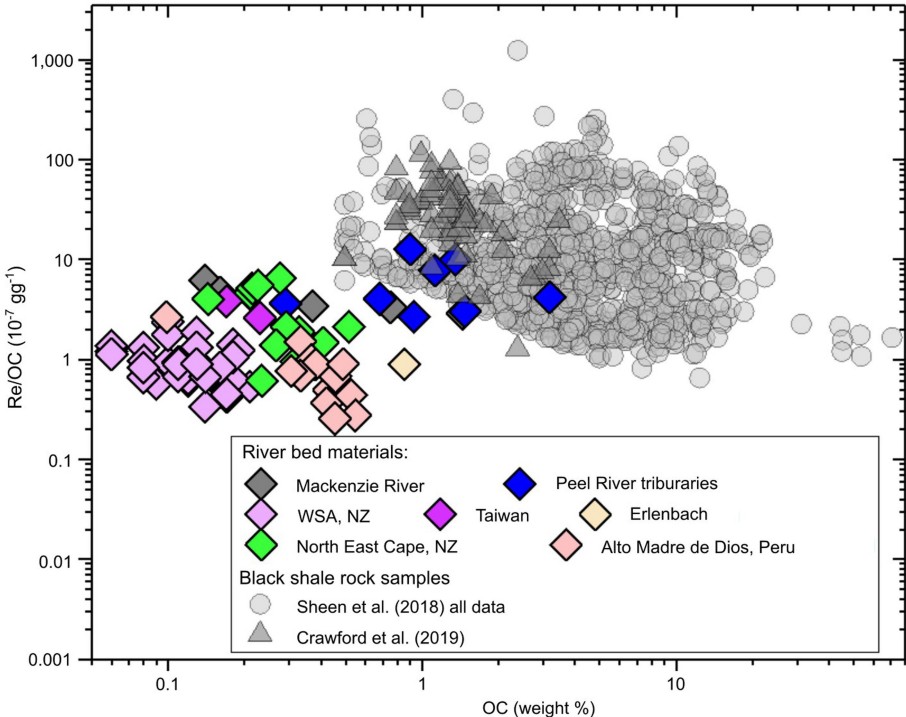

**Extended Data Fig. 2 | Rhenium (Re) to organic carbon (OC) concentration ratio ($10^{-7}$ g g$^{-1}$) in published black shales[54,67] alongside those measured in riverbed materials (Supplementary Table 3).** Peel riverbed materials capture the input of OC from black shale lithologies, which in other basins are much less common.

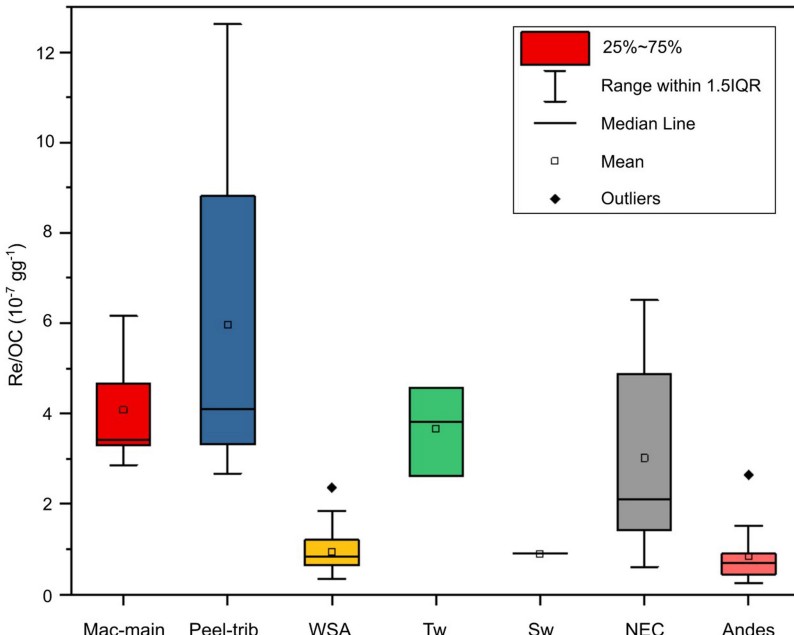

**Extended Data Fig. 3 | Rhenium (Re) to organic carbon (OC) concentration ratio ($10^{-7}$ g g$^{-1}$) of riverbed materials, organized by sample location (Supplementary Table 3).** Conservatively, the Monte Carlo analysis of OC$_{petro}$ oxidation rates from the dissolved rhenium proxy uniformly samples the complete measured range of Re/OC values presented here. Sample groups are as follows: Mac-main = Mackenzie River; Peel-trib = Peel river tributaries; WSA = Western Southern Alps, New Zealand; Tw = Taiwan rivers; Sw = Swiss rivers; NEC = North East Cape, New Zealand; Andes = Madre de Dios catchment.

a

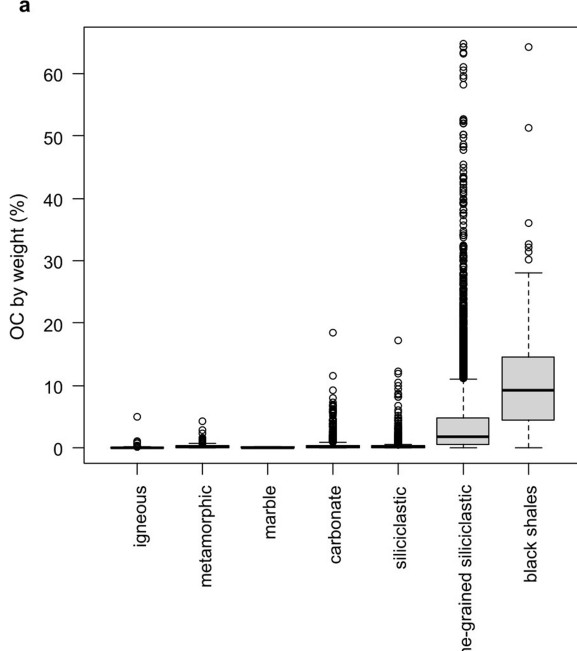

b

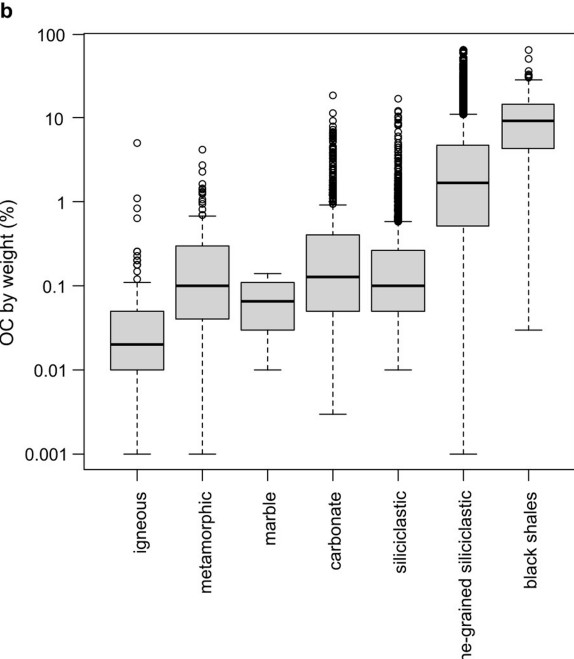

**Extended Data Fig. 4 | Organic carbon concentrations (OC in weight %) of lithological types in the USGS Rock Geochemical Database (a) and the same figure in log scale (b).** Black lines show median values, boxes show the interquartile range (IQR), whiskers show IQR + 1.5 × IQR, and outliers are shown as open symbols (although still included in the analysis).

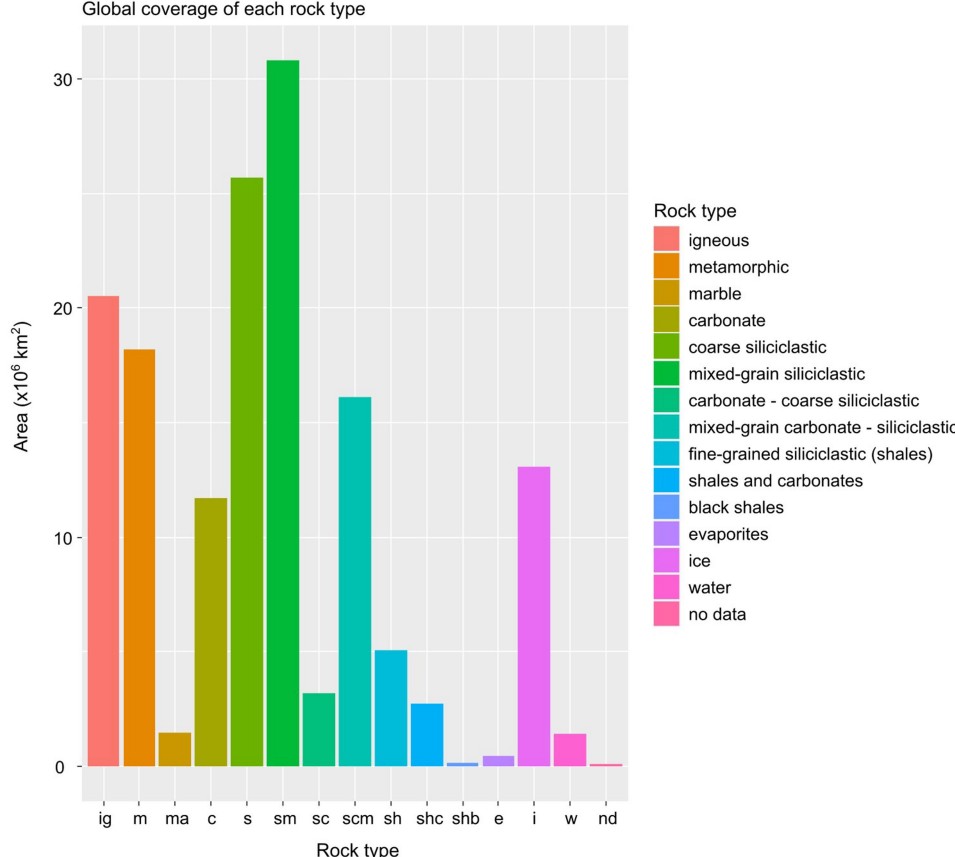

**Extended Data Fig. 5 | Total continental area (m²) of each mapped units from the GLiM model[13].** Mapped units include mixed units of siliciclastic lithologies with various grain-sizes (fine and coarse) and mixes of carbonate and siliciclastic lithologies[13].

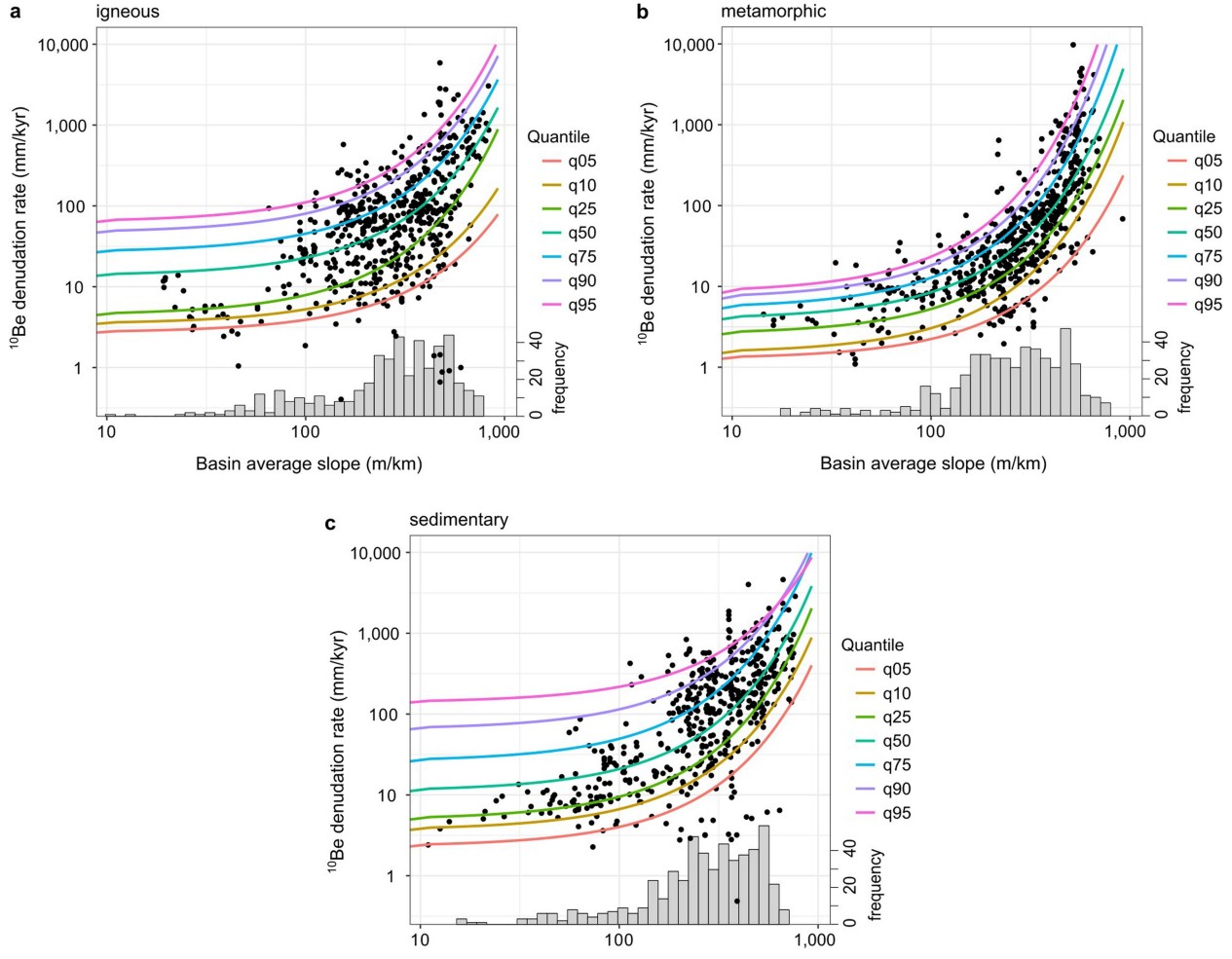

**Extended Data Fig. 6 | Cosmogenic isotope-derived denudation rates ($^{10}$Be erosion rate mm/kyr) from the OCTOPUS dataset[39] analysed here.** Sites are grouped by dominant lithology and the denudation rate as a function of basin average slope (m/km) returned by quantile regression for (**a**) igneous, (**b**) metamorphic, and (**c**) sedimentary dominated catchments. The quantiles corresponding to each grid cell slope value constitute cumulative density functions (CDFs) which can be sampled in each Monte Carlo simulation that quantifies global oxidative weathering rates (Extended Data Fig. 1).

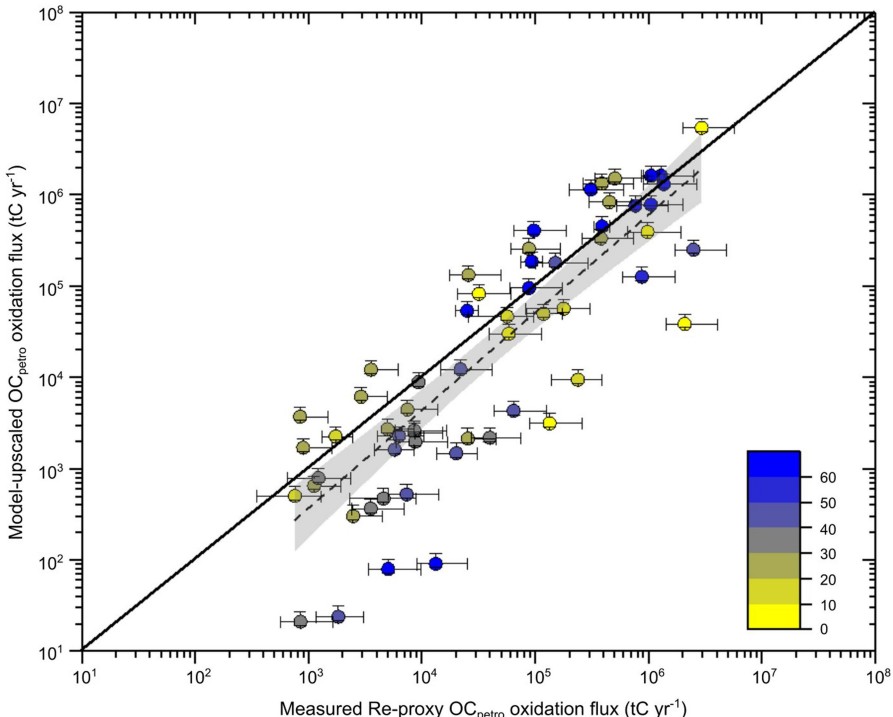

**Extended Data Fig. 7 | Rock organic carbon (OC_petro) oxidation rate (tC yr⁻¹) from the Re-proxy versus the model upscaled output for the global set of catchments used in this study (Supplementary Table 1).** Catchments are coloured by the latitude of the sampling point (0° to 80° N and S). Error bars represent uncertainty in Re-proxy values for OC_petro oxidation (see Methods sections headed 'Rhenium-based river catchment estimates of OC_petro oxidation' & 'OC_petro oxidation yields and uncertainties') and the resultant uncertainty in extrapolation model outputs. The dashed line and grey shading shows fit to the data (95% confidence intervals) and the solid black line shows 1:1 line.

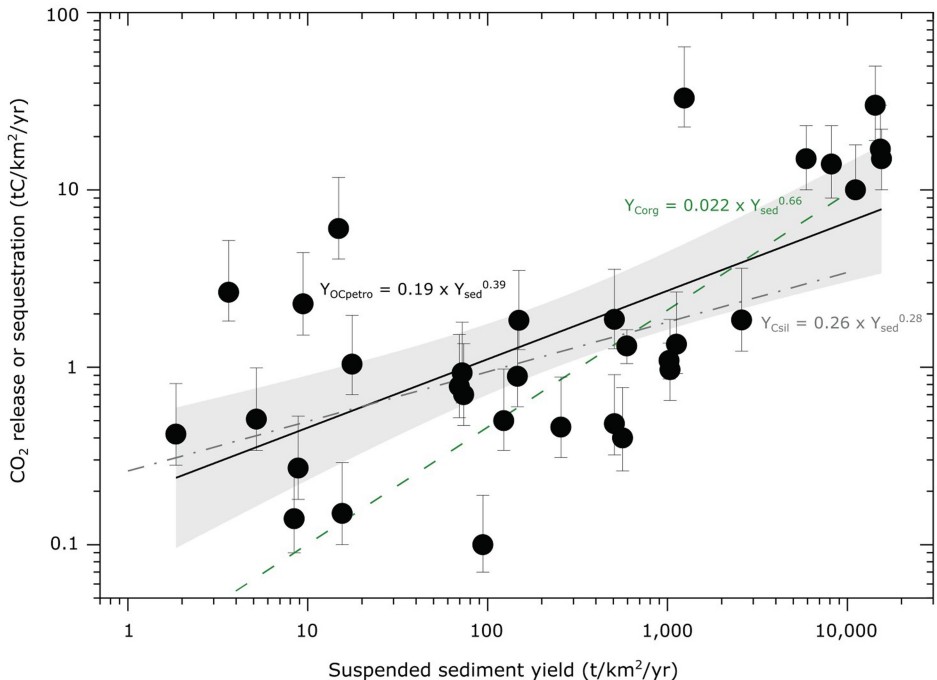

**Extended Data Fig. 8 | CO₂ release through OC_petro oxidation (black dots, this study; Y_OCpetro, black line) is more sensitive to sediment yield than CO₂ sequestration through silicate weathering (Y_Csil, grey dotted-dashed line)[4], whilst terrestrial biospheric OC burial (Y_Corg, green dashed line)[19] could be even more sensitive.** CO₂ sequestration trendlines are from Galy et al.[19] The black dots show CO₂ release through OC_petro oxidation for all the Re sample catchments which have suspended sediment yield data available from the Land2Sea database[68] (Supplementary Table 1) to allow comparison with the previous assessment of silicate weathering vs biospheric OC burial[19]. The exponent for $Y_{OCpetro}$ is $0.39 \pm 0.08$ with $r^2 = 0.43$, reflecting the scatter in data owing to the spatial dependence of OC_petro oxidation on rock OC content and denudation rates (Fig. 2). Therefore, comparisons based on the global trends in this plot, without considering their spatial context (see Fig. 2 for spatially explicit comparisons), should be done with care considering the low certainty.