## [Peer Review File · Nature]

Manuscript Title: Rock organic carbon oxidation CO₂ release offsets silicate weathering sink

Reviewer Comments & Author Rebuttals

Reviewer Reports on the Initial Version:

Referees' comments:

Referee #1 (Remarks to the Author):

The ms addresses a significant but poorly constrained process in the long-term global carbon cycle. The oxidation of ancient carbon (kerogen or "rock-organic carbon" or OC-petro) clearly takes place on a large scale but quantifying the magnitude of this overall process has been difficult. Most previous estimates were based on C isotope mass balance, but credible direct estimates have been lacking. The ms uses Re as a proxy for OC-petro oxidation and a simple form that depends on erosion rate and weathering intensity. The erosion rate term is obtained from a regression model of ¹⁰Be estimates of erosion vs. slope, applied to different lithologies. The weathering intensity is based on Re/OC ratios with corrections for sulfide and silicate derived Re. These relationships are placed in a spatial model to generate a global flux. Overall the approach seems well-rooted in previous work, some of which is from the same group over the past decade. A significant improvement is a data set on Re/OC in river sediments that are not sampling black shales, since black shales are locally important but globally a small fraction sediment.

The authors state (lines 124-125) that the variance in weathering intensity is < 2x, small compared to the variance in OC content of rocks and erosion rates. While the second part of that statement is very likely true, it's not clear from what is presented here that weathering intensity is mostly with a factor of 2. Perhaps that point can be made more explicitly, although I don't think a somewhat wider range will have that much impact on the overall result.

A strength of this ms is that estimates of OC-petro, lithology and denudation rate are placed in the context of other work on these issues. While it is not a guarantee it is useful to demonstrate that there is some 1st order agreement among different estimates. The quartile regression approach combined with monte Carlo sampling seems appropriate for the problem. One point of clarification – in several places in the ms and SI they mention using the "full" probability distribution. That seems vague – do they mean uniform sampling of the distribution? I think so but this can and should be stated using more standard terminology.

Figure 1: they summarize OC-petro in the upper 1 m as ≈ 1490 Gt C (with considerable uncertainty) and compare that to previous estimates and the soil C reservoir. For reference a recent estimate of OC in the upper 1 m of marine sediments is 2322 ± 75 (Atwood et al. FMS 2020).

Figure 2: It would be worthwhile in the SI to identify all the rivers in the compilation, as opposed to just some.

Figure 3: The caption needs to specify the units of the fluxes. The subduction flux number is not up to date, a better source and uncertainty would be Stolper et al. AJS 2021.

Overall I find this a well-done and valuable study that addresses a major gap in quantifying the C cycle. The global value they estimate ≈ 5.25 Tmol/yr is at least broadly consistent with a steady state value, where OC oxidation and OC burial are similar in magnitude, and what would be expected based on a $\delta^{13}\text{C}$ mass balance given what we know about carbonate burial (a better known number). Recent results support the notion of very small imbalances in the global C mass balance over the Cenozoic (Caves et al. 2016, Derry 2022) and the new estimate here is in accord

with that result. The finding that erosion and weathering processes in large watershed systems can be net sources or sinks of CO₂ when all the major sources are counted is both new on some sense but also agrees with detailed work such as Lupker et al. in the Narayani watershed (Himalaya).

I think this ms. can and should be published and needs only minor clarification. The Re proxy is as good a tool as we have at the moment for addressing this issue, and while the approach may have biases that are not yet fully recognized, the present study is clear about what they have done and represents a very good step forward.

Referee #2 (Remarks to the Author):

Zondervan et al. (Nature, submitted) use dissolved riverine Re concentrations and Re/Corg relationships in crustal rocks to estimate the fraction of petrogenic organic carbon that is respired upon erosion of OC-bearing rocks. Dubin & Peucker-Ehrenbrink (2015) have shown that a very significant fraction of the crustal Re is tied up in sediments enriched in organic matter. The authors argue that the Re-based approach illuminates a previously poorly quantified source of CO₂ to the atmosphere that counteracts CO₂ drawdown via weathering of silicate minerals. They further argue that CO₂ release via weathering of petrogenic organic matter exceeds CO₂ drawdown via silicate weathering in high-erosion environments, i.e. in times of more intense uplift and erosion, making orogenic periods in Earth history sources rather than sinks of atmospheric CO₂.

This topic is of great interest to the geoscience community and fitting for a high-impact journal such as Nature. The authors have compiled a large amount of relevant data and used the best available digital lithologic maps to model the global impact of petrogenic OC respiration. As such the manuscript is the logical continuation of a decade of research by Dr. Hilton and his team. Here, they are pushing the available evidence to the limit.

I think the authors' model is incomplete. While a fraction of weathering petrogenic OC is respired and adds CO₂ back to the atmosphere when petrogenic OC-bearing rocks erode, the other – refractory – fraction is transported to long-term sedimentary storage reservoirs and thus is transferred from one long-term reservoir to another. In addition, a fraction of biospheric OC is transported as POC to a long-term sedimentary storage reservoir. This comparatively large fraction (twice as large as the respired petrogenic OC fraction) is also sensitive to contributions from high erosion environments and responds with greater sensitivity to increased erosion than silicate weathering does, provided Galy et al. are correct in their assessment. Contributions from silicate weathering, respiration of petrogenic OC, and eroded biospheric OC transported as part of the suspended load have to be considered together, because these three fluxes are all sensitive to changing denudation rates, yet presumably with different sensitivities. It is therefore not clear to me whether high-erosion areas are net sources or sinks of CO₂. Can the authors' model provide conclusive insights into this combined balance, considering the uncertainties involved?

It should not be forgotten that there is an additional flux term that is not mentioned in the manuscript (unless it is subsumed under the volcanic flux) – CO₂ released from metamorphic reactions (not volcanism) in orogenic belts (see Kerrick & Caldeira, 1998; ¹⁸18 t/km²/yr). For the New England metamorphic belt Kerrick & Caldeira estimate an integrated C flux of ⁴⁰40 Mt C/yr – that seems significant compared to the other carbon fluxes. If I recalculate the Evans et al. (2008) data for the 100-times smaller Nepal Himalaya orogenic belt, I obtain an estimate of ^{0.16}0.16 Mt C/yr. Both independent estimates compare quite well and seem significant compared to the organic carbon fluxes.

Neglecting metamorphic fluxes for a moment, the authors' figure 3B picture all of the above processes. However, in figures 3C and D only the "Rock organic-carbon oxidation" flux scales with uplift and erosion. The flux labeled "Erosion and burial of biospheric organic carbon" also scales with uplift and erosion, apparently even more sensitively than silicate weathering, and it is the

larger of the two fluxes. It is unclear to me if the "Rock organic-carbon oxidation" flux scales even more sensitively to changes in uplift/erosion than the "erosion and burial of biospheric organic carbon" flux does. If it does, it could compensate, or even overcompensate, the increased organic carbon burial in times of increased suspended sediment fluxes. Unless we know this, I do not think we can state definitively which of the counteracting fluxes "wins out", and thus whether increased uplift/erosion is a net source (Zondervan's argument) or sink (Galy's argument) of atmospheric CO₂.

There is another aspect the authors may want to evaluate: whether the assertion by Rahaman et al. (2015) that Miller et al. (2011) significantly overestimated the global non-anthropogenic Re flux has merit. Rahaman et al. base their assertion on the strong positive correlation of dissolved Re and K in rivers, rather than relying on the good correlation between Re and SO₄ as Miller et al. (2011) did. If correct, this could mean that anthropogenic Re contributions are much more pervasive than previously thought. Rahaman et al. argue that anthropogenic contributions account for up to 70% of the dissolved Re budget in Indian peninsular drainages. The difference is very significant, leading to an averaged unpolluted dissolved riverine Re concentration of 3 pM (Rahaman et al., 2015) compared to the global estimate of 11 pM according to Miller et al. (2011). If Rahaman et al. are correct it would be much harder to "see through" the largely contaminated dissolved Re signal and use the natural portion for the purpose the authors intend to. It would also make any flux estimates that are a function of the dissolved (natural) Re flux significantly smaller.

For consistency in the presentation of fluxes, I suggest to use averages with uncertainties throughout the manuscript, rather than a mix of such averages and flux ranges. For instance, CO₂ drawdown via silicate weathering should be reported as 94±25 Mt C/yr rather than 70-120 Mt C/yr. It would also be good to include the biospheric CO₂ sink fluxes estimated by Galy et al. (157+74-50 Mt C/yr from POC/sediment load correlations, 140+96-57 Mt C/yr from NPP estimates; both are similar to the 170 Mt C/yr estimate the authors use in Fig. 3B, from Hilton & West, 2020). Add to that the 43+61-25 Mt C/yr refractory petrogenic OC yields a total contribution of 200+135-75 Mt C/yr. Zondervan's estimate of 68 (not 63 as stated in the abstract)+18-6 Mt C/yr is consistent with the inference by Galy et al. that up to 50% of eroding (petrogenic) POC could be respired during erosion and transport, thus adding CO₂ to the atmosphere. That estimate is in general agreement with Zondervan's more precise new estimate.

Regarding the now well-known Re enrichment in organic-rich sediments, I suggest the authors cite contributions by those who did pioneering research that showed that Re has the highest enrichment factors in reducing marine sediments of all redox-sensitive trace elements (V, Mo, Cd, U, etc.), particularly Goldberg (1987), Calvert & Pedersen (1993), and Morford & Emerson (1999), rather than citing reference [26].

The second Result equation on page 15 should be Resil instead.

Caption, Fig. 3: add "and" between "volcanic CO₂ degassing" and "silicate weathering."

Overall, I think this well-written manuscript has the potential to become a Nature paper. The authors quantify a previously not well quantified source of CO₂ to the atmosphere. I would accept a revised version that illuminates the overall balance of ALL significant CO₂ fluxes to the ocean-atmosphere system.

Bernhard Peucker-Ehrenbrink (WHOI)

Author Rebuttals to Initial Comments:

Response to Reviewer #1

Thank you for your review of our paper. We have answered each of your points below.

Please note that line numbers referred to in the response are those in the document with tracked changes.

The ms addresses a significant but poorly constrained process in the long-term global carbon cycle. The oxidation of ancient carbon (kerogen or “rock-organic carbon” or OC-petro) clearly takes place on a large scale but quantifying the magnitude of this overall process has been difficult. Most previous estimates were based on C isotope mass balance, but credible direct estimates have been lacking. The ms uses Re as a proxy for OC-petro oxidation and a simple form that depends on erosion rate and weathering intensity. The erosion rate term is obtained from a regression model of ^{10}Be estimates of erosion vs. slope, applied to different lithologies. The weathering intensity is based on Re/OC ratios with corrections for sulfide and silicate derived Re. These relationships are placed in a spatial model to generate a global flux. Overall the approach seems well-rooted in previous work, some of which is from the same group over the past decade. A significant improvement is a data set on Re/OC in river sediments that are not sampling black shales, since black shales are locally important but globally a small fraction sediment.

The reviewer sees this manuscript as a valuable contribution, and asks for some minor clarifications below. We have acted on these requests in the paragraphs and detailed comments below.

The authors state (lines 124-125) that the variance in weathering intensity is $< 2x$, small compared to the variance in OC content of rocks and erosion rates. While the second part of that statement is very likely true, it's not clear from what is presented here that weathering intensity is mostly with a factor of 2. Perhaps that point can be made more explicitly, although I don't think a somewhat wider range will have that much impact on the overall result.

Whilst the numbers presented in lines 123-124 imply that weathering intensity is mostly with a factor of 2, we agree with the author that the important point to get across is that the spatial variance is small compared to the variance in OC content of rocks and erosion rates.

To address this point, we have modified the text in lines 124-125:

Lines 131 – 135: “Weathering intensity χ has been shown to vary between low values of 0.2 in highly erosive settings⁷ and very high values of 0.98 in slow denudation settings⁸ with most falling in a range of 0.6 – 0.9^{7,30,31}. Thus, χ presents less than a factor of two a significantly smaller variance across environments in contrast to denudation rates and $[OC_{\text{petro}}]$ which vary spatially by more than four orders of magnitude.”

We leave the readers to judge any further interpretation of the numbers we present (i.e. if most variance falls within a factors of 2 or 3 etc..) for themselves.

A strength of this ms is that estimates of OC-petro, lithology and denudation rate are placed in the context of other work on these issues. While it is not a guarantee it is useful to demonstrate that there is some 1st order agreement among different estimates. The quartile regression approach combined with monte Carlo sampling seems appropriate for the problem. One point of clarification – in several places in the ms and SI they mention using the “full” probability distribution. That seems vague – do they mean uniform sampling of the distribution? I think so but this can and should be stated using more standard terminology.

We appreciate the reviewer’s positive comments about placing our estimates in the context of other work, and our use of the Monte Carlo sampling. The reviewer here asks us to use standard terminology where we refer to sampling values in Monte Carlo simulations. We agree with the author that “uniform sampling” describes our approach well and unambiguously.

To clarify, we changed these lines:

Lines 614 – 619: “To conservatively quantify uncertainty in the range of OC_{petro} oxidation rates from dissolved Re data we randomly sample perform a Monte Carlo simulation the full in which we uniformly sample the entire range of measured (OC_{petro}/Re) values, from low values indicative of carbon-poor and/or metamorphic rocks $2.5 \times 10^{-8} \text{ g g}^{-1}$ towards 1.26×10^{-6} in catchments with higher OC in rocks (Table S3; Extended Data Fig.3).”

Lines 625 – 628: “A Monte Carlo uncertainty propagation is used on these variables, with 10,000 randomly selected combinations of input values (with “full” uniform sampling probability distributions i.e., equal likelihood of a value between a range) are used to estimate $J_{OC_{\text{petro-ox}}}$ for each basin.”

Lines 906 – 909 (caption Extended Data Fig.3): Conservatively, Tthe full measured range is used in the Monte Carlo analysis of OC_{petro} oxidation rates from the dissolved rhenium proxy uniformly samples the complete measured range of Re/OC values presented here.

Figure 1: they summarize OC-petro in the upper 1 m as $\approx 1490 \text{ Gt C}$ (with considerable uncertainty) and compare that to previous estimates and the soil C reservoir. For reference a recent estimate of OC in the upper 1 m of marine sediments is 2322 ± 75 (Atwood et al. FMS 2020).

We thank the reviewer for an additional OC stock that the rock organic carbon stock is similar to in magnitude. We added this:

Lines 140 – 143: “This estimate is consistent with a global estimate of $1100 \text{ Gt } OC_{\text{petro}}$ within the first meter of sedimentary rocks¹⁴ and is of comparable magnitude to that of global soils ($2060 \pm 215 \text{ Gt OC}$)³⁹ and marine sediments ($2322 \pm 75 \text{ Gt OC}$)⁴⁰.”

Figure 2: It would be worthwhile in the SI to identify all the rivers in the compilation, as opposed to just some.

To address the reviewer's suggestion to identify the rivers in this plot, we have edited the labels so that each river that is mentioned in the manuscript is labelled on this Figure. In addition, we've included a source data file for this Figure for completion.

Figure 3: The caption needs to specify the units of the fluxes. The subduction flux number is not up to date, a better source and uncertainty would be Stolper et al. AJS 2021.

We agree with the reviewer that we should have added the units of fluxes, which we have now done. While we appreciate the reviewer's suggestion to use a number for subduction from Stolper et al. AJS 2021, we do not find a quantification of carbon subduction in this work. (Stolper, D.A., Higgins, J.A. and Derry, L.A., 2021. The role of the solid earth in regulating atmospheric O₂ levels. *American Journal of Science*, 321(10), pp.1381-1444.) As far as we're aware, the number in Plank & Manning Nature 2019 is the most up to date quantification of the flux we are after. (Plank, T. and Manning, C.E., 2019. Subducting carbon. *Nature*, 574(7778), pp.343-352.)

Overall I find this a well-done and valuable study that addresses a major gap in quantifying the C cycle. The global value they estimate ≈ 5.25 Tmol/yr is at least broadly consistent with a steady state value, where OC oxidation and OC burial are similar in magnitude, and what would be expected based on a $\delta^{13}\text{C}$ mass balance given what we know about carbonate burial (a better known number). Recent results support the notion of very small imbalances in the global C mass balance over the Cenozoic (Caves et al. 2016, Derry 2022) and the new estimate here is in accord with that result. The finding that erosion and weathering processes in large watershed systems can be net sources or sinks of CO₂ when all the major sources are counted is both new on some sense but also agrees with detailed work such as Lupker et al. in the Narayani watershed (Himalaya).

We are happy that the reviewer finds our results to make sense within the context of other related carbon cycle considerations.

I think this ms. can and should be published and needs only minor clarification. The Re proxy is as good a tool as we have at the moment for addressing this issue, and while the approach may have biases that are not yet fully recognized, the present study is clear about what they have done and represents a very good step forward.

We appreciate the reviewer's acknowledgement that our study presents a valuable contribution by addressing a significant but poorly constrained process in the long-term global carbon cycle.

Response to Reviewer #2 (Bernhard Peucker-Ehrenbrink, WHOI)

Thank you for your review of our paper. We have answered each of your points below. Please note that line numbers referred to in the response are those in the document with tracked changes.

Zondervan et al. (Nature, submitted) use dissolved riverine Re concentrations and Re/Corg relationships in crustal rocks to estimate the fraction of petrogenic organic carbon that is respired upon erosion of OC-bearing rocks. Dubin & Peucker-Ehrenbrink (2015) have shown that a very

significant fraction of the crustal Re is tied up in sediments enriched in organic matter. The authors argue that the Re-based approach illuminates a previously poorly quantified source of CO₂ to the atmosphere that counteracts CO₂ drawdown via weathering of silicate minerals. They further argue that CO₂ release via weathering of petrogenic organic matter exceeds CO₂ drawdown via silicate weathering in high-erosion environments, i.e. in times of more intense uplift and erosion, making orogenic periods in Earth history sources rather than sinks of atmospheric CO₂.

The reviewer emphasizes our use of the rhenium proxy in our approach to constraining a previously poorly quantified source of CO₂ to the atmosphere: rock organic carbon oxidation. The reviewer notes above, and in their review below, that we make two points:

1. That a) this newly constrained flux of rock organic carbon oxidation is a significant source of CO₂ from rock weathering which counteracts CO₂ drawdown via weathering of silicate minerals, and b) even exceeds CO₂ drawdown via silicate weathering in high-erosion environments. Therefore, rock weathering globally tends towards a geological source of CO₂ rather than a sink, with regions of high uplift and erosion contributing most to this source.
2. That therefore in times of more intense uplift and erosion, orogenic periods in Earth history are sources rather than sinks of atmospheric CO₂.

We fully agree that we present evidence to support point 1, which indeed is the title of this manuscript. Reviewer 1 understands this and find our work “a well-done and valuable study that addresses a major gap in quantifying the C cycle”.

Reviewer 2 uncovers another exciting implication of our results: our finding turning rock weathering from a sink to a source of CO₂ opens the question of whether periods in geological history of intensified uplift and erosion are those which tend to draw down or increase CO₂ levels in the atmosphere. However, a significant portion of reviewer 2's further comments can be understood in the context of a misunderstanding that we claim to fully conclude point 2 with our work alone (i.e. that we extrapolated our point 1 conclusion to point 2). In this manuscript we have focused on the net *rock weathering* carbon fluxes – i.e. those that occur when rocks are chemically weathered in contact with air, water and life. The full assessment of a complete global mountain building, erosion and weathering CO₂ budget, which includes burial of new organic matter in sedimentary deposits (and other fluxes such as metamorphic degassing during mountain building) is beyond the scope of our work. Nevertheless, by focusing on rock weathering we can highlight that periods in geological history of intensified uplift and erosion which enhance rock organic carbon oxidation (Figure 2) could see enhanced CO₂ release by rock weathering, and a less efficient silicate weathering sink. This new finding could drive the motivation for a canon of work on the dynamics of Earth's thermostat which may aim eventually to fully conclude on point 2.

Indeed, these other carbon cycle fluxes that are affected by mountain building which the reviewer discusses are part of our final discussion. These include the organic carbon burial in sediments which is driven by both marine and terrestrial-derived OC. This is also linked to continental weathering and erosion fluxes because: i) on long timescales, marine OC production can be limited by phosphorous supply from continental weathering; ii) supply of terrestrial particulate OC is strongly linked to erosion (Galy et al., 2015); and iii) preservation of OC in sediments is strongly linked to

sedimentation rate and the supply of clastic materials. Other factors play a role, including the protection of OC by mineral association (Hemingway et al., 2019) and the biogeochemical cycle of O_2 , which can govern burial at the seafloor. Thus, while the burial of marine and terrestrial biospheric organic carbon can “close” the geological C budget (Fig. 3), the OC burial CO_2 sink is not tied to the same drivers as silicate weathering and OC_{petro} weathering and oxidation. The dynamics of Earth’s thermostat thus need to be revisited to account for all of these major biospheric fluxes, and consider how their relative importance may have changed as life evolved and the OC stocks of sedimentary rocks increased (Galvez et al., 2020). While we can discuss these aspects, we must focus on our new understanding of the crucial rock weathering balance (silicate weathering vs rock organic weathering) in this manuscript. However, to address the reviewer’s request to illuminate our work in context of all significant CO_2 fluxes to the ocean-atmosphere system we have made additional references throughout the discussion. We detail these additions in response to the reviewer’s comments below.

We note that parts of our manuscript might not have been clear enough on our focus of the net rock weathering balance (weathering of silicate minerals vs rock organic matter), rather than a “complete” mountain building carbon budget (including erosion and marine burial of organic carbon and metamorphic degassing). In particular, we recognise that Figure 3 could have confused matters, as it could be misinterpreted that we were quantifying the latter. As such, we have redesigned the Figures 2 and 3 to better capture our new insight on the rock weathering CO_2 balance, while improving the clarity of the discussion on the wider set of processes that must be considered to understand net mountain building C budgets. We discuss these aspects in more detail in response to the reviewer’s comments below.

This topic is of great interest to the geoscience community and fitting for a high-impact journal such as Nature. The authors have compiled a large amount of relevant data and used the best available digital lithologic maps to model the global impact of petrogenic OC respiration. As such the manuscript is the logical continuation of a decade of research by Dr. Hilton and his team. Here, they are pushing the available evidence to the limit.

We are happy that reviewer 2 is in agreement with reviewer 1 that our work addressing a significant but poorly constrained process in the long-term global carbon cycle is fitting for publication in Nature. We are also happy that the author thinks our methodology of constraining the flux of petrogenic OC oxidation is a robust quantification of CO_2 release from rock weathering. We understand that the comment about pushing the available evidence to the limit comes from the misunderstanding that we are claiming point 2, detailed above. We therefore address this in response to the more detailed comments below.

I think the authors’ model is incomplete. While a fraction of weathering petrogenic OC is respired and adds CO_2 back to the atmosphere when petrogenic OC-bearing rocks erode, the other – refractory – fraction is transported to long-term sedimentary storage reservoirs and thus is transferred from one long-term reservoir to another.

We agree with the author that a fraction petrogenic OC is respired and the other transported unweathered. We do not agree that our model is incomplete in this manner, as we explicitly

constrain the flux of the weathered OC with the Re flux data (which quantifies the amount of petrogenic organic carbon that is respired). Indeed, further down the reviewer seems to acknowledge this: “Zondervan’s estimate of 68+18-6 Mt C/yr is consistent with the inference by Galy et al. that up to 50% of eroding (petrogenic) POC could be respired during erosion and transport, thus adding CO₂ to the atmosphere. That estimate in general agreement with Zondervan’s more precise new estimate.”

For clarity we further acknowledge the first order agreement between our respired flux estimate and those of the refractory transported flux in lines 176 – 185: “Using our spatial model, we estimate that oxidative weathering of OC_{petro} releases MtC yr^{-1} as CO₂ from the land-surface environment. [...] The flux is higher than an independent estimate of OC_{petro} erosion and river transport (i.e., the export of OC_{petro} that has not been weathered) of MtC yr^{-1} based on river solid load composition and flux¹⁹. While a direct comparison of these estimates is difficult based on their quantification from dissolved vs particulate river chemistry and flux, they suggest an average weathering intensity of OC_{petro} of ~60%, which is consistent with studies from large river basins¹⁹ and that intensities measured in soils^{8,44}.”

In addition, a fraction of biospheric OC is transported as POC to a long-term sedimentary storage reservoir. This comparatively large fraction (twice as large as the respired petrogenic OC fraction) is also sensitive to contributions from high erosion environments and responds with greater sensitivity to increased erosion than silicate weathering does, provided Galy et al. are correct in their assessment. Contributions from silicate weathering, respiration of petrogenic OC, and eroded biospheric OC transported as part of the suspended load have to be considered together, because these three fluxes are all sensitive to changing denudation rates, yet presumably with different sensitivities. It is therefore not clear to me whether high-erosion areas are net sources or sinks of CO₂. Can the authors’ model provide conclusive insights into this combined balance, considering the uncertainties involved?

We agree that biospheric OC is an important part of the puzzle, as shown in Figure 3 and the final part of the discussion. However, we disagree that they have to be considered together at this stage. Chemical weathering of silicate minerals and rock organic carbon happen in rock exposed at Earth’s surface and share a range of similar processes (supply of minerals, physical weathering and break up of mineral surfaces, supply of H₂O, residence time of fluids, residence time of rocks in weathering zones). In addition, the approaches applied in this paper to quantify rock organic carbon oxidation (dissolved river chemistry source partitioning and fluxes) are the same as those used to quantify silicate mineral weathering. It thus makes sense to directly compare these as we have done.

In contrast, the sink of CO₂ via erosion of biospheric OC is linked via denudation, but includes other important factors (spatial and temporal patterns of erosion and solid material export, mineral-organic carbon interactions in soils and floodplains, coastal sediment supply processes, sedimentation rate in marine settings, O₂ availability in marine settings). These fluxes are quantified using river sediment geochemistry, which is likely to have a different residence time on land than water and solutes, and assessments of longer-term burial offshore (over 1000s to millions of years). So we don’t think it appropriate to include it here in our catchment-by-catchment assessment of net rock weathering CO₂ balances. However, the total fluxes can be compared and discussed, as we do in the manuscript.

In order to make it clear that our manuscript focuses on a previously poorly quantified source of CO₂ to the atmosphere (and assesses the overall C balance of rock weathering; point 1 at the start of the review above) we edit the text to clarify this focus:

Lines 232-234: “We assess the net balance of rock weathering within major river basins (Fig. 2), using our OC_{petro} oxidation model and silicate weathering estimates¹⁰.”

Figure 2 caption: “Earth’s major river basins, their carbon sinks, and sources from silicate weathering¹⁰ and OC_{petro} weathering (this study), and their overall rock weathering budget based on these weathering fluxes.”

Lines 249 – 251: “Within uncertainties, rock weathering in about a third of the major river basins is a net source of CO₂ during chemical weathering after OC_{petro} oxidation is considered, even without accounting for downstream carbonate precipitation (Fig. 2a).”

Lines 261 – 266: “On the other hand, basins that where rock weathering is are the biggest net sinks of CO₂ do not necessarily lie at the extremes of low denudation and low OC_{petro} stocks. Whilst the tropical Congo and volcanics-dominated Godavari basins have low basin-average denudation rates and low OC_{petro} stocks, the biggest weathering sink, the Amazon, lies in the global middle range of denudation rates and OC_{petro} stocks (Fig. 2b-c).”

Lines 269 – 271: “A in a third of river basins weathering remains carbon neutral within uncertainty such as the volcanic-rich Columbia catchment.”

The wording in the introductory paragraph may have contained some ambiguity. To prevent duplication of introduction with the summary paragraph along editorial guidelines, this content is now contained solely within the summary paragraph (“abstract”). Here we emphasize our focus on the role of rock weathering during mountain building. Whether mountain building itself is overall a net source or sink of CO₂, including all other carbon fluxes such as modern biospheric organic carbon production and transport, is an interesting and very significant extended implication of our work. We come back to this implication in the discussion. While the abstract is already carefully focused on rock weathering, we make an additional edit to further clarify:

Lines 29 – 30: “Our results demonstrate that rock organic carbon is far from inert and causes weathering in regions to be either net sources or sinks of CO₂ from weathering, ...”

It should not be forgotten that there is an additional flux term that is not mentioned in the manuscript (unless it is subsumed under the volcanic flux) – CO₂ released from metamorphic reactions (not volcanism) in orogenic belts (see Kerrick & Caldeira, 1998; "18 t/km²/yr). For the New England metamorphic belt Kerrick & Caldeira estimate an integrated C flux of "40 Mt C/yr – that seems significant compared to the other carbon fluxes. If I recalculate the Evans et al. (2008) data for the 100-times smaller Nepal Himalaya orogenic belt, I obtain an estimate of "0.16 Mt C/yr. Both independent estimates compare quite well and seem significant compared to the organic carbon fluxes.

We agree with the author that metamorphic decarbonation and diffuse CO₂ release in subduction zones are an important contribution to the solid-Earth degassing flux (which is the flux quantified by the cited study). We had been incomplete in our wording of this flux, as indeed volcanic release is only part of this flux. To address the reviewer's comment we have changed the wording in the manuscript from volcanic degassing to solid-Earth degassing to be complete:

Lines 275 – 277: “Volcanic Solid-Earth degassing associated with volcanoes and diffuse release from subduction zones is responsible for 79 ± 9 MtC yr⁻¹ released into the atmosphere (Fig. 3a; ⁴⁹)”

Figure 3: changed wording on the figure to “Solid-Earth degassing (incl. volcanoes)” + caption: “a global balance between volcanic solid-Earth CO₂ degassing”

In addition, we agree that metamorphism in orogenic belts including those that are not subduction zones generate CO₂ fluxes that could further support our point that CO₂ sources might be more significant in mountain belts than previously considered. To address this point, in combination with other points raised by the reviewer, we have expanded our consideration of all carbon fluxes to the ocean-atmosphere system (lines 275 - 294). Without an available global estimation of metamorphically generated CO₂, it is difficult for us to attempt a quantitative comparison with the other globally constrained fluxes we cite here. In lines 275 – 280 we further acknowledge this flux:

“Volcanic Solid-Earth degassing associated with volcanoes and diffuse release from subduction zones is responsible for 79 ± 9 MtC yr⁻¹ released into the atmosphere (Fig. 3A; ⁴⁹), while any additional (non-subduction) global CO₂ release during orogenic metamorphism and sulfide oxidation and marine inorganic C uptake during seafloor weathering are more poorly constrained³.”

Neglecting metamorphic fluxes for a moment, the authors' figure 3B picture all of the above processes. However, in figures 3C and D only the “Rock organic-carbon oxidation” flux scales with uplift and erosion. The flux labeled “Erosion and burial of biospheric organic carbon” also scales with uplift and erosion, apparently even more sensitively than silicate weathering, and it is the larger of the two fluxes. It is unclear to me if the “Rock organic-carbon oxidation” flux scales even more sensitively to changes in uplift/erosion than the “erosion and burial of biospheric organic carbon” flux does. If it does, it could compensate, or even overcompensate, the increased organic carbon burial in times of increased suspended sediment fluxes. Unless we know this, I do not think we can state definitively which of the counteracting fluxes “wins out”, and thus whether increased uplift/erosion is a net source (Zondervan's argument) or sink (Galy's argument) of atmospheric CO₂.

Reviewer 2 here interprets Figure 3 to say that increased uplift/erosion is a net source of atmospheric CO₂. However, the points of panels c and d is to say that rock weathering can be net sources or sinks depending on the uplift/erosion rate. This is an observation which adds to the overall finding that rock organic carbon oxidation appears to negate silicate weathering at the current distribution of denudation and bedrock geology. We addressed this potential misunderstanding with modifications throughout the written manuscript, in response to the previous time this issue was raised in this review (see above – two comments earlier). Here, we address the reviewer's comments on Figure 3 and the accompanying discussion.

To address the potential for misunderstanding and address the reviewer's comment, we have modified Figure 3 by moving panels c and d, which can be better understood by examining the balance between rock weathering processes under varying uplift/erosion in Figure 2. Consequently, the panels belong to the discussion around Figure 2 where already in the submitted manuscript these panels were cited.

Figure 3 (panels a and b) addresses the implications for the geological carbon cycle overall which is a major theme of Reviewer #2s comment. Here, our message is that the rock organic carbon oxidation flux makes rock weathering a net source of CO₂, leading to the implication of a large dependence of Earth's long-term climate stability on other sinks, such as the erosion and burial of organic biospheric carbon. Consequently, as the reviewer notes, we need to revisit the dynamics of Earth's thermostat: in particular how organic carbon burial into sedimentary rocks responds to changing environmental conditions such as increased uplift and erosion or changes in climate is not as well constrained yet (as this depends not just on terrestrial biospheric carbon transport but also marine organic carbon production and overall preservation and burial in sedimentary rocks). We now carefully lay out the implications of our work as portrayed in Figure 3 in the accompanying discussion:

Lines 280 – 294:

“As our results show that the weathering of OC_{petro} negates silicate weathering in the long-term carbon cycle, a large additional sink of CO₂ is needed. This may be provided by burial of organic matter in sediments, which could contribute as much as 170 MtC yr⁻¹ (Figure 3b; ⁵⁰). In addition, as OC_{petro} fluxes can overtake silicate weathering during periods of more intense uplift and erosion (Fig. 2), the question whether orogenic periods in Earth history are sources or sinks of atmospheric CO₂ has now become an open question which depends on factors such as the transport of terrestrial biospheric carbon to oceans (_____ MtC yr⁻¹) which is also more sensitive to physical erosion rates than silicate weathering¹⁹. However, while the subsequent burial of terrestrial and marine biospheric organic carbon can apparently close the geological C budget (Fig. 3B; ⁵⁰), the OC burial CO₂ sink may not be tied to the same tectonic and climatic drivers in the same way as silicate weathering, OC_{petro} oxidation, and terrestrial biospheric carbon transport⁴⁴. The dynamics of Earth's thermostat thus need to be revisited to account for variation in all these fluxes and consider how their relative importance may have changed as life evolved and the OC stocks of sedimentary rocks have increased^{3,22}.”

There is another aspect the authors may want to evaluate: whether the assertion by Rahaman et al. (2015) that Miller et al. (2011) significantly overestimated the global non-anthropogenic Re flux has merit. Rahaman et al. base their assertion on the strong positive correlation of dissolved Re and K in rivers, rather than relying on the good correlation between Re and SO₄ as Miller et al. (2011) did. If correct, this could mean that anthropogenic Re contributions are much more pervasive than previously thought. Rahaman et al. argue that anthropogenic contributions account for up to 70% of the dissolved Re budget in Indian peninsular drainages. The difference is very significant, leading to an averaged unpolluted dissolved riverine Re concentration of 3 pM (Rahaman et al., 2015) compared to the global estimate of ~11 pM according to Miller et al. (2011). If Rahaman et al. are

correct it would be much harder to “see through” the largely contaminated dissolved Re signal and use the natural portion for the purpose the authors intend to. It would also make any flux estimates that are a function of the dissolved (natural) Re flux significantly smaller.

We agree with the reviewer that the anthropogenic flux of Re is an important consideration to make, especially in our approach of using this flux to constrain the natural process of rock organic carbon weathering. We considered this to be an important issue to address and make our constraints robust, and we had consequently thought through and addressed this issue extensively even in our initial manuscript. This may not have come through very clearly to the reviewer, so to address this we have strengthened the communication of our methodology.

We thank the reviewer for pointing us to this study which we have now cited. The results of Rahaman suggest that Indian peninsular drainages contain significant anthropogenic Re. They use concentrations of Re, and relationships between Re and other solutes (cations, SO_4^{2-} , K^+) to make this case, finding a signal of Re enhancement in a selection of rivers where anthropogenic Re is invoked (Godvari) and/or black shale weathering is locally important (Yamuna River). However, the same study discusses that Re in the Himalayan catchments and the mainstem Ganges and Brahmaputra rivers are mostly from natural sources. We do not use any Indian peninsular catchments to drive our model. Here we want to add a note that whilst we compare our spatial model output of the Godavari to the model output for silicate weathering by Moon et al. (2014), we do not train our model on Re fluxes from the Godavari.

In line with the reviewer's comments, we had already excluded rivers in the Miller 2011 dataset which are known to have significant anthropogenic Re pollution (see Table S1). Whilst it is likely that our training catchment dataset of Re data will still contain some pollution, our model is conservative in its predictions of rock organic carbon oxidation (and thus not apparently biased by any systematic anthropogenic Re input) for two reasons:

1. Our conversion of Re fluxes to OC_{petro} oxidation is conservative because in our Monte Carlo simulation rather than sampling the empirical probability distribution we uniformly sample the range of Re/OC ratios starting at the lowest measured Re/OC ratio. This leads to error bars within our estimates that are conservatively large.
2. More importantly, we compare our model predictions with Re-derived measures of rock organic carbon oxidation in our catchments (Extended Data Fig. 7). In contrast to the larger catchments included in the Miller 2011 dataset which account for 56% of our overall dataset, the rest of our dataset contains mostly catchments that were carefully selected by the authors of these studies to minimise human disturbance (Table S1 for citations) and measured very low Re concentrations in precipitation (Horan et al., 2017; Hilton et al., 2021). Most of these are smaller mountain catchments with high uplift and erosion. These upland areas also contribute the majority of global rock organic carbon oxidation in our model: lines 199 – 203. Overall, Extended Data Fig. 7 shows that the upland mountain areas which have minimal anthropogenic impacts on Re (which are typically smaller basins with lower total fluxes) have model fluxes that are lower than measured: i.e. in these regions which contribute most of the global flux the model underpredicts rather than overpredicts. We therefore have confidence that the effect of any residual anthropogenic Re pollution in

our training dataset is unlikely to result in a significant overprediction of global OC_{petro} flux estimates.

To communicate these points more clearly throughout the manuscript we modified and added the following text:

Lines 552– 568: “In locations with significant local sources of fossil fuel combustion (e.g., coal-fired power plants or steel works), rainwater can contain concentrations of Re that approach those of river water^{8,52}, whereas locations that have minimal impacts from local pollution sources have Re concentrations in rainwater that are below detection^{25,30}. In the large river dataset⁹ some large rivers are noted to have markedly increased Re concentrations and fluxes, with the conclusion that this was due to anthropogenic Re inputs. A study of Re across Indian catchments suggests that whilst Re in Himalayan catchments and the mainstem Ganges and Brahmaputra behave conservatively, peninsular lower relief catchments with denser populations and industrial activity suggest anthropogenic inputs⁵³. For this study's purposes to quantify weathering reactions, we only use Himalayan rivers and the mainstem Ganges and Brahmaputra in India and we have not infurther excluded Re data from the Danube, Mississippi, and Yangtze rivers in from our analysis. Our addition of catchment Re data to the Miller dataset includes a large contribution of small upland catchments with higher average erosion rates, where the authors of these studies selected sites with minimal human disturbance (Table S1). We further consider the role of anthropogenic Re in our model results in section 3.”

Lines 769 – 783: “Results of model vs Re-predicted OC_{petro} oxidation fluxes help us to assess the potential for anthropogenic Re input to impact our estimates (Extended Data Fig. 7). We have considered anthropogenic Re inputs by removing three large river basins from a previous compilation⁹ and by adding carefully selected river catchment sites to our Re dataset (Section 1.2). In addition, our conversion of Re fluxes to OC_{petro} oxidation is conservative because we uniformly sample the range of Re/OC ratios starting at the lowest measured Re/OC ratio (Section 1.2.2). This leads to error bars within our estimates that are conservatively large. Most notably, the model outputs of OC_{petro} oxidation vs the Re-estimated fluxes for each basin (Extended Data Fig. 7) show a tendency for the model to underpredict smaller catchments more than larger catchments. Our confidence in the weathering signal from Re in the small upland catchments is highest, and the upland, high erosion rate regions that these catchments sample contribute a dominant proportion of the global OC_{petro} flux in our model. Consequently, whilst we cannot completely deconvolve the effect of anthropogenic Re in our constraints, we have confidence that the effect is unlikely to result in a significant overprediction of global OC_{petro} flux estimates.”

For consistency in the presentation of fluxes, I suggest to use averages with uncertainties throughout the manuscript, rather than a mix of such averages and flux ranges. For instance, CO₂ drawdown via silicate weathering should be reported as 94 ± 25 Mt C/yr rather than 70-120 Mt C/yr.

We agree with the reviewer that it is better to report the numbers consistently in the suggested format. To address this we changed this throughout the manuscript:

Line 214: “global net CO₂ uptake by terrestrial silicate weathering ($70\ 12094 \pm 25$ MtC yr⁻¹)¹⁰.”

Line 217: “resulting in a net transfer of $35\ 6047 \pm 12$ MtC year⁻¹.”

Lines 275 – 277: “Volcanic Solid-Earth degassing associated with volcanoes and diffuse release from metamorphism in subduction zones is responsible for $70\ 10079 \pm 9$ MtC yr⁻¹ released into the atmosphere (Fig. 3a; ⁴⁹)”

We also tracked these format changes into the number reporting on Figure 3.

It would also be good to include the biospheric CO₂ sink fluxes estimated by Galy et al. (157+74-50 Mt C/yr from POC/sediment load correlations, 140+96-57 Mt C/yr from NPP estimates; both are similar to the ~170 Mt C/yr estimate the authors use in Fig. 3B, from Hilton & West, 2020).

We added the flux estimate of terrestrial biospheric carbon export from Galy et al. in lines 283 – 288:

Lines 280 – 294:

“As our results show that the weathering of OC_{petro} negates silicate weathering in the long-term carbon cycle, a large additional sink of CO₂ is needed. This may be provided by burial of organic matter in ocean sediments, which could contribute as much as 170 MtC yr⁻¹ (Figure 3B; ⁵⁰). In addition, as OC_{petro} fluxes likely overtake silicate weathering during periods of more intense uplift and erosion, the question whether orogenic periods in Earth history are sources or sinks of atmospheric CO₂ has now become an open question which depends on factors such as the transport of terrestrial biospheric carbon to oceans (_____ MtC yr⁻¹) which is also more sensitive to physical erosion rates than silicate weathering¹⁹. However, while the subsequent burial of terrestrial and marine biospheric organic carbon can apparently close the geological C budget (Fig. 3B; ⁵⁰), the OC burial CO₂ sink may not be tied to the same tectonic and climatic drivers in the same way as silicate weathering, OC_{petro} oxidation, and terrestrial biospheric carbon transport⁴⁴. The dynamics of Earth’s thermostat thus need to be revisited to account for variation in all these fluxes and consider how their relative importance may have changed as life evolved and the OC stocks of sedimentary rocks have increased⁵¹.”

Add to that the 43+61-25 Mt C/yr refractory petrogenic OC yields a total contribution of 200+135-75 Mt C/yr. Zondervan’s estimate of 68 (not 63 as stated in the abstract)+18-6 Mt C/yr is consistent with the inference by Galy et al. that up to 50% of eroding (petrogenic) POC could be respired during erosion and transport, thus adding CO₂ to the atmosphere. That estimate in general agreement with Zondervan’s more precise new estimate.

We are happy to hear the reviewer seems to find consistency of our global average weathering intensity estimates and those inferred from field studies of the Ganges-Brahmaputra and Amazon, as it is useful to demonstrate that there is some 1st order agreement among different estimates. We already discuss these themes in the paper.

We cite the refractory petrogenic OC yield constrained by Galy et al. in line 181, and now also mention the agreement with the Galy et al estimation of weathering intensity:

Lines 176 – 185: “Using our spatial model, we estimate that oxidative weathering of OC_{petro} releases MtC yr^{-1} as CO_2 from the land-surface environment. [...] The flux is higher than an independent estimate of OC_{petro} erosion and river transport (i.e., the export of OC_{petro} that has not been weathered) of MtC yr^{-1} based on river solid load composition and flux¹⁹. While a direct comparison of these estimates is difficult based on their quantification from dissolved vs particulate river chemistry and flux, they suggest an average weathering intensity of OC_{petro} of ~60%, which is consistent with studies from large river basins¹⁹ and that intensities measured in soils^{8,44}.”

We also thank the reviewer for noticing the typo in the summary paragraph (“abstract”), which we have now corrected:

Line 26: “We find a CO_2 release of MtC yr^{-1} megatons of carbon from weathering of organic carbon in near-surface rocks annually”

Regarding the now well-known Re enrichment in organic-rich sediments, I suggest the authors cite contributions by those who did pioneering research that showed that Re has the highest enrichment factors in reducing marine sediments of all redox-sensitive trace elements (V, Mo, Cd, U, etc.), particularly Goldberg (1987), Calvert & Pedersen (1993), and Morford & Emerson (1999), rather than citing reference [26].

We agree that reference 26 only captures a very small amount of the work that has been done on the topic, we did so for its work explicitly linking Re to the organic compounds present in sedimentary rocks. References 27 and 28 also capture some of the pioneering work on Re in rivers and its behaviour during weathering. Unfortunately we cannot cite all these works within the reference count, but we have rationalised the reference list and now include Morford et al., 2012, “Rhenium geochemical cycling: Insights from continental margins”, *Chemical Geology* 324–325 (2012) 73–86, as this provides an overview of accumulated knowledge that includes some of these workers, and is directly relevant to the theme of where Re resides and accumulates in marine sediments.

The second Resulf equation on page 15 should be Resil instead.
We have corrected this typo. Thanks to the reviewer for spotting this!

Caption, Fig. 3: add “and” between “volcanic CO_2 degassing” and “silicate weathering.”
We have also corrected this typo. Thanks to the reviewer for spotting this!

Overall, I think this well-written manuscript has the potential to become a Nature paper. The authors quantify a previously not well quantified source of CO_2 to the atmosphere. I would accept a revised version that illuminates the overall balance of ALL significant CO_2 fluxes to the ocean-atmosphere system. We thank the reviewer for their positive and constructive review. With the revisions made in response to this review we hope that we have now carefully illuminated the implications of our results on the overall balance of all CO_2 fluxes to the ocean-atmosphere system that the reviewer raised.

Reviewer Reports on the First Revision:

Referees' comments:

Referee #1 (Remarks to the Author):

The revised version of Zondervan et al. reasonably addresses issues raised in the first round of review. In addition to clarifying some of the model assumptions, it now includes some additional discussion of the global C cycle balance which raises an interesting dilemma. This ms has the first data-driven estimate of the oxidation rate of ancient organic carbon (C-petro); previously this number was simply assumed to be similar to long term organic carbon burial fluxes.

The ms discusses the strong localization of the OC-petro source. I suggest this could be made plainer. Lines 177-180 seem more complex than needed. Why not just say that the two drivers in the model are 1) OC petro content, which is spatially variable (i.e. a lot of rocks have very little) and erosion rate which is very spatially variable. The part about "kinetic limitation links temperature and runoff ..." seems pretty beside the point because they previously explain that those factors are weak predictors of OC petro oxidation. So why bring it up again, it just makes things confusing? This paragraph could usefully and easily be simplified and would be better focused.

As the ms summarizes, we now have estimates of long term CO₂ sources including volcanism and arc-region metamorphism ($\approx 6.6 \text{ Tmol yr}^{-1}$ or 79 Mton, e.g. Plank & Manning 2019, and others), and now from C-petro oxidation (5.7 Tmol yr^{-1} (68 Mton), this ms).

We also have estimates of sinks from silicate weathering (7.0 to 7.8 Tmol yr^{-1} or $\approx 90 \text{ Mton/yr}$, after Gaillardet et al 1999 and Moon et al. 2019). We have various estimates of Corg burial (a sink) that span a wide range (as discussed in Burdige 2006), including two recent ones (Smith et al., 2015, working from Hedges and Keil 1995) and Li et al. 2023 (who quote Dunne et al 2012 but this is not a global estimate). These estimates of Corg burial are ≈ 10 to 14 Tmol yr^{-1} (130 – 170 Mton C yr⁻¹)

If the Corg burial rate estimates are in fact valid for long term fluxes, there is a serious mass balance problem because C sinks would appear to exceed C sources by a large and implausible margin (50 to 80%), but this is not an issue with the current ms who are not responsible for the Corg burial estimates. If instead the classic $\delta^{13}\text{C}$ mass balance is used to estimate Corg burial from Ccarb burial the source-sink imbalance is within uncertainty of zero. The Corg burial sink estimated with $\delta^{13}\text{C}$ (ca. 6 Tmol yr^{-1} , or 72 Mton yr^{-1}) is not dissimilar to the C-petro source given here, which is a hint that this may a reasonable set of assumptions that close the C cycle budget. It's hard to make the C cycle work if the large Corg burial fluxes are close to correct, unless we are way off on weathering and degassing fluxes, which seems unlikely on several grounds. The degassing source and weathering sink are estimated independently from one another and appear to be a) moderately well constrained and within uncertainty of each other, suggesting that the inorganic part of the C cycle is close to being in balance. That would also be true of the organic part of the C cycle if the new OC-petro source given here is about right and the $\delta^{13}\text{C}$ based OC burial sink value is about right. This would all be nice and I think could be correct but this is not yet clear.

The notion advanced here that orogenic events could be either sources or sinks of CO₂ has come up before (Evans et al. G3 2008; Spence and Telmer GCA 2006; Torres et al 2014; Horan et al. 2019; Marki et al. Nat Geo 2021) but is well stated here with the additional new insight that they gained with Re-OC systematics. So I would keep that idea, while perhaps giving a shout out to

previous authors.

There are a couple places where things are slightly unclear and minor editing can fix them – this doesn't need further review, in my opinion.

Lines 90-91 has some text about a chemical weathering model, but there isn't one in this ms. The authors simply use the results from Moon et al. 2019. I found this confusing and unnecessary. I would remove the sentence that begins "Chemical weathering is modelled (sic)..." as well as the unnecessary references #32,33,34. I don't see at all what this adds to this ms, and it gives the reader the impression that there is a silicate weathering model coming up. There is not.

Line 245: I would replace the word "negates" with "balances" or better "approximately balances".

Line 253: I disagree – as pointed out above a large OC burial flux as proposed by Smith (based on Hedges) does not balance if the other terms (degassing, silicate weathering, OC petro) are about right. In that scenario, with the other fluxes quoted here, the sources are considerably less than the sinks. So while I think the intent of the sentence was to say that that the OC-petro source could be offset by the OC burial sink (and that's fine) the published estimates they cite have "too much" OC burial and so overshoot, i.e. it more than compensates the OC-petro source. This is not the authors responsibility to fix in this ms, but it does point to the mass balance problem that we appear to have. This can be clarified.

Overall this is a nice contribution that provides a new and independent constraint on an important long term C cycle flux term we previously mostly had to guess at. That's a very good result. There are just a few things to clarify. Keep the story straightforward and on track.

Lou Derry

Referee #2 (Remarks to the Author):

This is a significantly improved version that more accurately describes what Zondervan et al. do address and what not. I dispense with the summary of key results, the originality and significance that I have already commented on in my original review. Below are a number of issues to consider for the final version:

The title is not as accurate as I would like it to be, because the effects of silicate weathering are not entirely negated. For instance, the release of nutrients (N, P, Si) from silicate weathering still affect river and ocean biogeochemistry. Something like "Global rock organic carbon oxidation releases more CO₂ than silicate weathering consumes" would be more accurate.

Line 27/28 – Remember that the biosphere is also a BIG part of the overall CO₂ balance.

Line 519 – The flux estimate is not higher within the stated uncertainties.

Line 530 – Grant et al. (2023, already cited by the authors) is relevant in the context of soil carbon turnover timescales.

Line 606 – The authors are careful in correcting CO₂ drawdown by silicate weathering for short-term effects by reducing the overall flux to $\frac{1}{2}$. However, the value cited in ref. 10 (94+₋₂₅ Mt C/yr) does not include volcanic islands. If those are included, ref. 10 estimates a CO₂ drawdown of 143 Mt C/yr (see ref. 10 abstract). I also wonder whether similar corrections need to be made to the estimated release of CO₂ by rock organic carbon oxidation. Not all of the rock organic carbon that is lost in the weathering process is lost as CO₂. Otherwise Petsch et al. (2000, Nature) would

not have detected ^{14}C -dead carbon in microbes consuming rock-derived organic carbon. Some weathered organic carbon may be released as dissolved organic carbon to streams and rivers with a range of potential fates, including adsorption onto suspended particles and ultimate burial as sedimentary organic matter. It would be fair to point out that only a fraction of the organic carbon lost from rocks enters the aquatic-atmospheric system as CO_2 .

Fig. 2a/b show the relation of the release of C from oxidation of rock-derived organic carbon to that consumed (long-term) by silicate weathering as a function of denudation rate. I urge the authors to make those figures directly comparable to Galy et al. (2015, extended data figure 2) to enable a direct comparison between the carbon balance of silicate weathering, rock-derived organic carbon oxidation and biogenic organic carbon sequestration as part of the river suspended matter flux. The authors are so close to enable this direct comparison. My interpretation of Fig. 2b is that the crossover between silicate weathering domination and rock-organic-carbon oxidation (at around 25 mm/kyr denudation) corresponds to $\sim 60 \text{ t/sqkm/yr}$ in the way Galy et al. plot their data. However, the slightly steeper slope of the rock-organic-carbon oxidation relationship compared to the silicate weathering relationship is subtle, as indicated by the linear scale used on the y-axis of Fig. 2b. If the authors don't calculate the exponent of the log-log CO_2 sequestration yield vs. sediment yield (Galy et al., 2015) relationship someone else will and then be able to compare all three contributions quantitatively. I urge the authors to push their data just a little bit more – THEY ARE SO CLOSE TO MAKING THIS IMPORTANT COMPARISON.

Fig. 2d/e – The figures 2d and 2e need to be relabeled in the figure caption because 2d = source, 2e = sink.

I do not think the carbon output ($79 \pm 9 \text{ Mt/yr}$) in ref. 49 includes metamorphic release of CO_2 in collisional orogens mentioned in my original review. The "intraplate" carbon output in ref. 49 refers to "intraplate volcanoes" and "calderas and geothermal systems".

Line 744 – It is worth remembering that export and burial of organic matter in the oceans is also a function of nutrients liberated by silicate (and phosphate) mineral weathering.

Ref. 15 – subscript O_2 (molecular oxygen) is missing.

Ref. 36/38 – subscript CO_2 is missing

Line 1249 – Brahmaputra is spelled incorrectly.

Once finalized, this will be an important contribution to the field.

Bernhard Peucker-Ehrenbrink (WHOI)

Author Rebuttals to First Revision:

Response to Referee #1 (Lou Derry, Cornell)

We have answered each of the referee's points below.

Please note that line numbers referred to in the response are those in the document with tracked changes.

The revised version of Zondervan et al. reasonably addresses issues raised in the first round of review. In addition to clarifying some of the model assumptions, it now includes some additional discussion of the global C cycle balance which raises an interesting dilemma. This ms has the first data-driven estimate of the oxidation rate of ancient organic carbon (C-petro); previously this number was simply assumed to be similar to long term organic carbon burial fluxes.

The reviewer appreciates our handling of the first round of review, and asks for some minor edits below. We have acted on these requests in the paragraphs and detailed comments below.

The ms discusses the strong localization of the OC-petro source. I suggest this could be made plainer. Lines 177-180 seem more complex than needed. Why not just say that the two drivers in the model are 1) OC petro content, which is spatially variable (i.e. a lot of rocks have very little) and erosion rate which is very spatially variable. The part about "kinetic limitation links temperature and runoff ... " seems pretty beside the point because they previously explain that those factors are weak predictors of OC petro oxidation. So why bring it up again, it just makes things confusing? This paragraph could usefully and easily be simplified and would be better focused.

To address the referee's suggestion to simplify this paragraph we removed lines 177-180 as suggested. Indeed our paragraph now focuses just on the spatial variability of OC_{petro} content and relief, which drives denudation:

Lines 173 – 188: "Across the land surface, OC_{petro} weathering is relatively focused (Fig. 1d), with variations in rock type and relief, which drive OC_{petro} content and denudation respectively, determining the magnitude of OC_{petro} oxidation and CO₂ release. Large regions of the African continent have lower OC stocks in bedrock and have lower relief, which together limit OC weathering. In contrast, higher OC_{petro} oxidation rates are estimated for northern latitudes, where OC-rich rock and high-relief landscapes are more prevalent. Overall, 10% of the Earth surface with the highest OC_{petro} oxidation rates account for 60% of the global flux in our model. The world average rate is 0.5 tC km⁻² yr⁻¹, hotspots (10 times world average) and hyperactive areas (all areas >5 times world average) are responsible for 32% and 44% of CO₂ emissions, while representing only 1.2 % and 3% of ice-free terrestrial land area, respectively. OC_{petro} weathering rates in our model are more spatially concentrated than a 1-km resolution spatial model of silicate weathering⁴⁵, where hotspots (0.51% by area) and hyperactive areas (2.9% by area) accounted for 8.6% and 19.6% of total CO₂-consumption. This outcome appears reasonable because OC_{petro} is less common spatially than silicate minerals and react faster^{3,25}. Hotspots of OC_{petro} oxidation could be places where kinetic limitation links temperature and runoff to CO₂ release^{36,37}, linking climate to a geological CO₂ emission."

As the ms summarizes, we now have estimates of long term CO₂ sources including volcanism and arc-region metamorphism ($\approx 6.6 \text{ Tmol yr}^{-1}$ or 79 Mton, e.g. Plank & Manning 2019, and others), and now from C-petro oxidation (5.7 Tmol yr^{-1} (68 Mton), this ms). We also have estimates of sinks from silicate weathering (7.0 to 7.8 Tmol yr⁻¹ or $\approx 90 \text{ Mton/yr}$, after Gaillardet et al 1999 and Moon et al. 2019). We have various estimates of Corg burial (a sink) that span a wide range (as discussed in Burdige 2006), including two recent ones (Smith et al., 2015, working from Hedges and Keil 1995) and Li et al. 2023 (who quote Dunne et al 2012 but this is not a global estimate). These estimates of Corg burial are ≈ 10 to 14 Tmol yr^{-1} (130 – 170 Mton C yr⁻¹). If the Corg burial rate estimates are in fact valid for long term fluxes, there is a serious mass balance problem because C sinks would appear to exceed C sources by a large and implausible margin (50 to 80%), but this is not an issue with the current ms who are not responsible for the Corg burial estimates. If instead the classic d13C mass balance is used to estimate Corg burial from Ccarb burial the source-sink imbalance is within uncertainty of zero. The Corg burial sink estimated with d13C (ca. 6 Tmol yr^{-1} , or 72 Mton yr⁻¹) is not dissimilar to the C-petro source given here, which is a hint that this may a reasonable set of assumptions that close the C cycle budget. It's hard to make the C cycle work if the large Corg burial fluxes are close to correct, unless we are way off on weathering and degassing fluxes, which seems unlikely on several grounds. The degassing source and weathering sink are estimated independently from one another and appear to be a) moderately well constrained and within uncertainty of each other, suggesting that the inorganic part of the C cycle is close to being in balance. That would also be true of the organic part of the C cycle if the new OC-petro source given here is about right and the d13C based OC burial sink value is about right. This would all be nice and I think could be correct but this is not yet clear.

We thank the referee for this thoughtful note. We are also excited by the implications of our work for those seeking to estimate organic carbon burial, and the impact on global isotopic mass balance. The referee asks us to change some wording in the discussion in his last comment below, which we address there.

The notion advanced here that orogenic events could be either sources or sinks of CO₂ has come up before (Evans et al. G3 2008; Spence and Telmer GCA 2006; Torres et al 2014; Horan et al. 2019; Marki et al. Nat Geo 2021) but is well stated here with the additional new insight that they gained with Re-OC systematics. So I would keep that idea, while perhaps giving a shout out to previous authors.

To address the reviewer's point that the question of whether orogenic events could be either sources or sinks of CO₂ has been asked before, we acknowledge this in lines 261 – 264, where we mention this. We have added a reference to the recent (2020) review on Mountains, erosion and the carbon cycle (Nature Reviews Earth & Environment), and references which are not included in this review, such as Evans et al. 2008 and Marki et al. 2021). We were able to do this and stay within the limit for references through the edit suggested by this referee in the next comment below.

Lines 263 -266: "In addition, as OC_{petro} fluxes can overtake silicate weathering during periods of more intense uplift and erosion (Fig. 2; Extended Data Fig. 8), the question whether orogenic periods in Earth history are sources or sinks of atmospheric CO₂ is now an reopened question_{3,31,49,50}."

There are a couple places where things are slightly unclear and minor editing can fix them – this doesn't need further review, in my opinion.

Lines 90-91 has some text about a chemical weathering model, but there isn't one in this ms. The authors simply use the results from Moon et al. 2019. I found this confusing and unnecessary. I would remove the sentence that begins "Chemical weathering is modelled (sic)..." as well as the unnecessary references #32,33,34. I don't see at all what this adds to this ms, and it gives the reader the impression that there is a silicate weathering model coming up. There is not.

To address the redundancy which may lead to confusion as pointed out by the referee, we have removed this sentence (line 90-91) and associated references. This has given us the space to add the important references which the referee suggested in his comment above.

Lines 91 – 97: "Our model incorporates topographic and lithological factors to estimate OC_{petro} stocks, denudation rates and oxidative weathering rates, and is calibrated using our Re-proxy compilation (Table S1). Chemical weathering is modelled as a tradeoff between mineral supply to weathering reactions³², and the kinetics of fluid rock interaction^{33,34}. Unlike silicate weathering, which quickly becomes kinetically limited with increasing mineral supply by denudation³⁵, OC_{petro} weathering appears to be predominately a supply-limited process⁸."

Line 245: I would replace the word "negates" with "balances" or better "approximately balances".

To address with both referees' suggestions to replace the word "negate" we have used the word "offsets" instead. The term "offsets" in the revised manuscript captures the idea of one process counteracting the other, without implying that they are exactly equal, which addresses Referee 1's concern.

Lines 258 – 260: "As our results show that the weathering of OC_{petro} negates offsets silicate weathering in the long-term carbon cycle, a large additional sink of CO_2 is needed."

Line 253: I disagree – as pointed out above a large OC burial flux as proposed by Smith (based on Hedges) does not balance if the other terms (degassing, silicate weathering, OC petro) are about right. In that scenario, with the other fluxes quoted here, the sources are considerably less than the sinks. So while I think the intent of the sentence was to say that that the OC-petro source could be offset by the OC burial sink (and that's fine) the published estimates they cite have "too much" OC burial and so overshoot, i.e. it more than compensates the OC-petro source. This is not the authors responsibility to fix in this ms, but it does point to the mass balance problem that we appear to have. This can be clarified.

To address the referee's sharp perception, we fixed wording of our sentence to reflect our intent. Indeed, rather than exactly matching, the OC_{petro} source could be offset by the OC burial sink.

Lines 266 – 270: "However, while the subsequent burial of terrestrial and marine biospheric organic carbon can apparently close offset or even overcompensate OC_{petro} oxidation in the geological C

budget (Fig. 3b; ⁴⁸), the OC burial CO₂ sink may not be tied to tectonic and climatic drivers in the same way as silicate weathering and OC_{petro} oxidation⁴².”

Overall this is a nice contribution that provides a new and independent constraint on an important long term C cycle flux term we previously mostly had to guess at. That's a very good result. There are just a few things to clarify. Keep the story straightforward and on track.

Lou Derry

We appreciate the reviewer's acknowledgement that our study presents a valuable contribution, and his comments which helped us edit for further clarity.

Response to Referee #2 (Bernhard Peucker-Ehrenbrink, WHOI)

We have answered each of the referee's points below.

Please note that line numbers referred to in the response are those in the document with tracked changes.

This is a significantly improved version that more accurately describes what Zondervan et al. do address and what not. I dispense with the summary of key results, the originality and significance that I have already commented on in my original review. Below are a number of issues to consider for the final version:

The title is not as accurate as I would like it to be, because the effects of silicate weathering are not entirely negated. For instance, the release of nutrients (N, P, Si) from silicate weathering still affect river and ocean biogeochemistry. Something like “Global rock organic carbon oxidation releases more CO₂ than silicate weathering consumes” would be more accurate.

To address with both referees' suggestions to replace the word “negate” we have used the word “offsets” instead. Referee 2 mentions that the title should be more accurate and reflect that the effects of silicate weathering are not entirely negated. The revised title addresses this by using “offsets” the “silicate weathering sink” of CO₂ instead of “negates” silicate weathering. The new title conveys that the CO₂ release from rock organic carbon oxidation counteracts the CO₂ sink from silicate weathering, without implying complete cancellation of all its effects.

New title: “Rock organic carbon oxidation CO₂ release offsets silicate weathering sink”

Line 27/28 – Remember that the biosphere is also a BIG part of the overall CO₂ balance.

We agree with the reviewer's point that the biosphere should be included in this sentence of the summary paragraph, to reflect the balance of carbon fluxes we explore.

Lines 26 – 30: “Our results demonstrate that rock organic carbon is far from inert and causes weathering in regions to be either net sources or sinks of CO₂, calling into question how erosion and weathering drive the long-term carbon cycle and contribute to the fine balance of carbon fluxes

between the atmosphere, biosphere and lithosphere^{2,11}.”

Line 519 – The flux estimate is not higher within the stated uncertainties.

The line numbering in this comment and the ones following do not seem to correspond to the line numbering of the submitted document. We have however located the corresponding sentences and address these below.

To address the referee’s concern of the uncertainties being left out of this statement, we have altered the sentence to add this nuance:

Lines 155 – 158: “The best estimate of the oxidative weathering flux is higher than an independent estimate of OC_{petro} erosion and river transport (i.e., the export of OC_{petro} that has not been weathered) of MtC yr⁻¹ based on river solid load composition and flux¹⁹, even though the uncertainties overlap.”

Line 530 – Grant et al. (2023, already cited by the authors) is relevant in the context of soil carbon turnover timescales.

To address this point by the referee we added a line connecting our results to the findings by Grant et al. (2023):

Lines 171 – 172: “Consequently, the input of OC_{petro} into soils affect soil residence time estimations and can lead to an underestimation of soil carbon exchange fluxes with the atmosphere²⁰.”

Line 606 – The authors are careful in correcting CO₂ drawdown by silicate weathering for short-term effects by reducing the overall flux to $1/2$. However, the value cited in ref. 10 (94+₋₂₅ Mt C/yr) does not include volcanic islands. If those are included, ref. 10 estimates a CO₂ drawdown of 143 Mt C/yr (see ref. 10 abstract). I also wonder whether similar corrections need to be made to the estimated release of CO₂ by rock organic carbon oxidation. Not all of the rock organic carbon that is lost in the weathering process is lost as CO₂. Otherwise Petsch et al. (2000, Nature) would not have detected ¹⁴C-dead carbon in microbes consuming rock-derived organic carbon. Some weathered organic carbon may be released as dissolved organic carbon to streams and rivers with a range of potential fates, including adsorption onto suspended particles and ultimate burial as sedimentary organic matter. It would be fair to point out that only a fraction of the organic carbon lost from rocks enters the aquatic-atmospheric system as CO₂.

To address the referee’s suggestion to point out the fate of OC_{petro} released from the lithosphere by oxidation, we added the following line:

Lines 64 – 66: “In addition, studies tracking the fate of carbon released from the lithosphere during rock organic carbon weathering have found it can: directly enter the atmosphere as CO₂^{30,31} or first dissolve as inorganic carbon in water³², and some can be incorporated into microbial biomass⁸.”

To make this comparison with the silicate weathering flux more comparable (i.e. the flux of CO₂ between the lithosphere and the atmosphere/hydrosphere/biosphere, we reworded lines 192 – 195. In addition, to address the referee's point that the Moon et al study gives two estimates, we now reflect this in the numbers cited in the text and Figure 3.

Lines 192 – 195: “Silicate weathering involves dissolved and gaseous CO₂ uptake through bicarbonate production and the release of calcium ions, which precipitate as marine carbonate rocks⁴. The resultant total transfer of carbon from the atmosphere to the lithosphere by silicate weathering is 47 - 72 MtC year⁻¹. However, over the timescale of carbon storage in the ocean, half of the CO₂ absorbed by silicate weathering at the reaction site is offset by CO₂ release during carbonate precipitation and burial in oceans downstream⁴, resulting in a net transfer of 47 ± 12 MtC year⁻¹.”

Fig. 2a/b show the relation of the release of C from oxidation of rock-derived organic carbon to that consumed (long-term) by silicate weathering as a function of denudation rate. I urge the authors to make those figures directly comparable to Galy et al. (2015, extended data figure 2) to enable a direct comparison between the carbon balance of silicate weathering, rock-derived organic carbon oxidation and biogenic organic carbon sequestration as part of the river suspended matter flux. The authors are so close to enable this direct comparison. My interpretation of Fig. 2b is that the crossover between silicate weathering domination and rock-organic-carbon oxidation (at around 25 mm/kyr denudation) corresponds to ~60 t/sqkm/yr in the way Galy et al. plot their data. However, the slightly steeper slope of the rock-organic-carbon oxidation relationship compared to the silicate weathering relationship is subtle, as indicated by the linear scale used on the y-axis of Fig. 2b. If the authors don't calculate the exponent of the log-log CO₂ sequestration yield vs. sediment yield (Galy et al., 2015) relationship someone else will and then be able to compare all three contributions quantitatively. I urge the authors to push their data just a little bit more – THEY ARE SO CLOSE TO MAKING THIS IMPORTANT COMPARISON.

The referee suggests we do two things:

- 1) Supply a plot of data that compares to Extended Data Figure 2 by Galy et al. 2015. (<https://www.nature.com/articles/nature14400/figures/5>)
 - 2) Facilitate the interpretation of a crossover between silicate weathering domination and rock-organic-carbon oxidation.
-
- 1) To address the first suggestion by the referee to directly compare our OC_{petro} oxidation constraints to the plots of Galy et al. 2015, we added Extended Data Figure 8. We agree that it is worth plotting a comparable figure that allows assessment of the sensitivity of all three fluxes to suspended sediment yield. This is why we used the Re-derived catchment data where there is available suspended sediment yield data available from the same source as used by Galy et al. 2015 (the Land2Sea database⁶⁸). This data allows us to plot in the same x and y range as their Extended Data Figure 2b. The sediment yield data is now also available in Table S1.

Extended Data Figure 8.

CO₂ release through OC_{petro} oxidation (black dots, this study; Y_{OCpetro}, black line) is more sensitive to sediment yield than CO₂ sequestration through silicate weathering (Y_{Csil}, grey dotted-dashed line)⁴, whilst terrestrial biospheric OC burial (Y_{Corg}, green dashed line)¹⁹ could be even more sensitive. CO₂ sequestration trendlines are from Galy *et al.*¹⁹. The black dots show CO₂ release through OC_{petro} oxidation for all the Re sample catchments which have suspended sediment yield data available from the Land2Sea database⁶⁸ (Supplementary Table S1) to allow comparison with the previous assessment of silicate weathering vs biospheric OC burial¹⁹. The exponent for Y_{OCpetro} is 0.38 ± 0.08 with r² = 0.43, reflecting the scatter in data owing to the spatial dependence of OC_{petro} oxidation on rock OC content and denudation rates (Fig. 2). Therefore, comparisons based on the global trends in this plot, without considering their spatial context (see Fig. 2 for spatially explicit comparisons), should be done with care considering the low certainty.

Lines 263 – 266: “In addition, as OC_{petro} fluxes can overtake silicate weathering during periods of more intense uplift and erosion (Fig. 2; Extended Data Fig. 8), the question whether orogenic periods in Earth history are sources or sinks of atmospheric CO₂ is now a reopened question^{3,31,49,50}.”

Lines 268 – 274: “A global comparison of catchment-scale OC_{petro} oxidation yields and estimated terrestrial biospheric OC burial (Extended Data Figure 8) suggest the However, while the subsequent OC burial of terrestrial and marine biospheric organic carbon can apparently offset or even overcompensate CO₂ release from OC_{petro} oxidation. This understanding persists when the additional marine OC burial sink in sediment is factored into global flux estimates (Fig. 3b; 48).”

2) In Figure 2, we use our spatial modelling capability to assess net weathering fluxes for major river basins on Earth. This is the most robust approach for such an assessment since the OC_{petro} oxidation flux is highly spatially dependent on rock OC_{petro} content and its overlap with denudation rates, as detailed and discussed in the manuscript. To address the suggestion by the referee to aid in the interpretation of a crossover between silicate weathering domination and rock-organic-carbon oxidation, we added a trendline and arrows to Fig. 2b to aid in this interpretation, and added the estimate in the figure caption:

Figure 2 caption: "...Net weathering balance vs basin average denudation (cross-over at $\sim 30 \text{ mm kyr}^{-1}$)"

In addition, Extended Data Figure 8 should now help in interpreting a similar crossover in units of sediment yield.

Fig. 2d/e – The figures 2d and 2e need to be relabeled in the figure caption because 2d = source, 2e = sink.

We thank the referee for their perceptive observation. We have corrected this typo in Figure 2's caption.

Lines 224 - 225: "...regions where rock weathering is (d) a sink source or (e) source sink of CO_2 ."

I do not think the carbon output ($79 \pm 9 \text{ Mt/yr}$) in ref. 49 includes metamorphic release of CO_2 in collisional orogens mentioned in my original review. The "intraplate" carbon output in ref. 49 refers to "intraplate volcanoes" and "calderas and geothermal systems".

We agree with the referee that global estimates do not include carbon output from metamorphic release in collisional orogens. The referee mentioned some studies with regional estimates in his previous review. While Figure 3 cites the currently available global estimate of $79 \pm 9 \text{ Mt/yr}$, in the accompanying text we address this:

Lines 254 – 258: "Solid-Earth degassing associated with volcanoes and diffuse release from metamorphism in subduction zones is responsible for $79 \pm 9 \text{ MtC yr}^{-1}$ released into the atmosphere (Fig. 3a; ⁴⁸), while any additional (non-subduction) global CO_2 release during orogenic metamorphism and sulfide oxidation and inorganic C uptake during seafloor weathering are more poorly constrained³."

Line 744 – It is worth remembering that export and burial of organic matter in the oceans is also a function of nutrients liberated by silicate (and phosphate) mineral weathering.

We agree with the referee that OC burial of terrestrial and marine biospheric organic carbon have various drivers which are more intricate than the (simpler) tectonic and climatic drivers of rock weathering, but to discuss these is out of the scope our manuscript. We currently cover this in the following sentence:

Lines 266 – 270: “However, while the subsequent burial of terrestrial and marine biospheric organic carbon can apparently offset or even overcompensate OC_{petro} oxidation in the geological C budget (Fig. 3b; ⁴⁸), the OC burial CO_2 sink may not be tied to tectonic and climatic drivers in the same way as silicate weathering and OC_{petro} oxidation⁴².”

Ref. 15 – subscript O₂ (molecular oxygen) is missing.

We’ve fixed this typo.

Ref. 36/38 – subscript CO₂ is missing

We’ve fixed this typo.

Line 1249 – Brahmaputra is spelled incorrectly.

We’ve fixed this typo.

Once finalized, this will be an important contribution to the field.

Bernhard Peucker-Ehrenbrink (WHOI)

Reviewer Reports on the Second Revision:

Referees' comments:

Referee #2 (Remarks to the Author):

Zondervan et al. have responded to my latest criticism by including the extended figure 8 that shows how the three components of the surficial carbon cycle respond to varying denudation rates. The data make sense: oxidation of petrogenic carbon in rocks has a slightly steeper slope than CO₂ drawdown by silicate weathering as both respond to essentially the same forcings. Contributions from biospheric carbon have steeper slope because they also respond to primary productivity.

The authors hold on to their lower estimate (I consider this “incomplete”) of CO₂ drawdown by silicate weathering (94 MtC per year) but also include the (what I consider “complete) figure of 143 MtC year) by citing the full range (line 201). This also applies to the corrected total transfer of 47-72 MtC per year. I think the higher number is better supported by the existing data.

There is a subscript missing in Reference 4 “CO₂”.

Overall, I think this paper is ready for publication. It will be a fine contribution to the field.

Bernhard Peucker-Ehrenbrink.